# Quantum error correction with silicon spin qubits

Kenta Takeda[1 ✉], Akito Noiri[1], Takashi Nakajima[1], Takashi Kobayashi[2] & Seigo Tarucha[1,2 ✉]

Future large-scale quantum computers will rely on quantum error correction (QEC) to protect the fragile quantum information during computation[1,2]. Among the possible candidate platforms for realizing quantum computing devices, the compatibility with mature nanofabrication technologies of silicon-based spin qubits offers promise to overcome the challenges in scaling up device sizes from the prototypes of today to large-scale computers[3–5]. Recent advances in silicon-based qubits have enabled the implementations of high-quality one-qubit and two-qubit systems[6–8]. However, the demonstration of QEC, which requires three or more coupled qubits[1], and involves a three-qubit gate[9–11] or measurement-based feedback, remains an open challenge. Here we demonstrate a three-qubit phase-correcting code in silicon, in which an encoded three-qubit state is protected against any phase-flip error on one of the three qubits. The correction to this encoded state is performed by a three-qubit conditional rotation, which we implement by an efficient single-step resonantly driven iToffoli gate. As expected, the error correction mitigates the errors owing to one-qubit phase-flip, as well as the intrinsic dephasing mainly owing to quasi-static phase noise. These results show successful implementation of QEC and the potential of a silicon-based platform for large-scale quantum computing.

Quantum computing takes advantage of quantum superposition and entanglement to accelerate the computational tasks[12,13]. However, these quantum properties are sensitive to decoherence errors owing to energy relaxation and dephasing. As the number of qubits increases and/or the computational tasks become more complex, the errors cause exponential reduction of the accuracy of computational results. QEC is a protocol to circumvent this problem by distributing the quantum information across a larger multiqubit entangled state so that the errors can be detected and corrected[14]. Its basic concept has been demonstrated in various platforms, such as nuclear magnetic resonance[9,15], trapped ions[10,16], nitrogen vacancy centres[17] and superconducting circuits[11,18,19], and has served as an important benchmark of the qubit systems. Silicon-based spin qubits have emerged as a qubit platform in the past decade, and there has been rapid progress in long coherence times[20,21], high-fidelity universal quantum gates[6–8], high-temperature operation[22,23] and generation of three-qubit entanglement[24,25].

Our three-qubit system (Fig. 1a) comprises one data qubit ($Q_2$) to be corrected and two ancilla qubits ($Q_1$ and $Q_3$). The sequence starts from encoding the data qubit state to a three-qubit entangled state. Then the phase-flip errors that occurred in the encoded state are mapped to the ancilla qubit states by the decoding. The original data qubit state can finally be restored by a correcting logic gate based on the ancilla qubit states. Most commonly, this correction can be performed by a projective measurement of ancilla qubits followed by a feedback quantum gate on the data qubit. However, this requires a capability to perform high-fidelity qubit measurement much faster than the coherence time, which is still challenging with spins in silicon. Although this measurement-based operation is a key component for

fault tolerance, in the case of three-qubit QEC, it can alternatively be performed by a multiqubit conditional qubit rotation. In this Article, we take this approach by using a three-qubit iToffoli gate, which coherently rotates the data qubit conditioned on the ancilla spin polarization. We synthesize a three-qubit phase-flip code and demonstrate that one-qubit phase-flip error can be corrected and the intrinsic ensemble spin dephasing can be mitigated.

Our sample is a gate-defined triple quantum dot in an isotopically natural silicon/silicon-germanium (Si/SiGe) heterostructure. Three layers of overlapping aluminium gates[26] are used to control the triple-dot confinement. A micro-magnet is fabricated on top of the aluminium gates to provide a local magnetic field gradient[27]. As schematically shown in Fig. 1b, we configure the gate voltages so that only one electron is confined under each of the plunger gates (P1, P2 and P3) and the inter-dot tunnel coupling is controlled by the barrier gates (B2 and B3). Measurement of the triple-dot charge configuration is performed by monitoring the conductance of the nearby charge sensor quantum dot using the radio-frequency reflectometry technique[28,29]. An in-plane external magnetic field of $B_{ext} = 0.607$ T is applied using a superconducting magnet. We use the Zeeman-split spin-1/2 states of the single electrons as our spin qubits (labelled $Q_1$, $Q_2$ and $Q_3$ in Fig. 1b,c). The Zeeman energy splitting (about 20 GHz) much larger than the thermal excitation energy (about 0.8 GHz or 40 mK) enables initialization and readout of the three-spin state by the combination of energy-selective tunnelling[30], shuttling[31] and controlled rotation (see Methods and Extended Data Fig. 1 for the full details of the sequence).

The single-qubit rotations are performed by applying resonant microwave pulses (see Methods and Extended Data Fig. 2). The

[1]Center for Emergent Matter Science (CEMS), RIKEN, Wako, Japan. [2]Center for Quantum Computing (RQC), RIKEN, Wako, Japan. ✉e-mail: kenta.takeda@riken.jp; tarucha@riken.jp

**a**

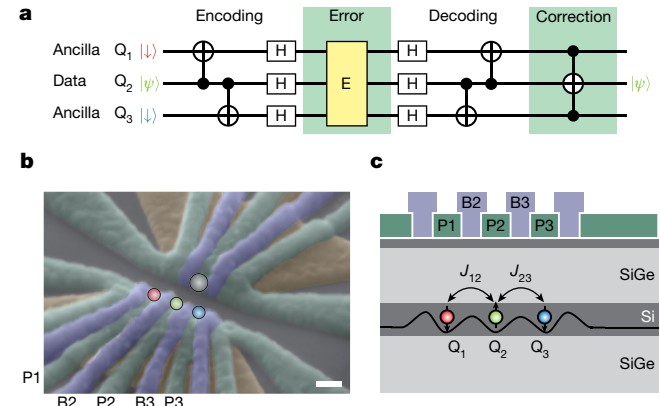

| Ancilla | $Q_1$ | $|\downarrow\rangle$ | Encoding | | | Error | | Decoding | | | Correction | | |
|---|---|---|---|---|---|---|---|---|---|---|---|---|---|
| Data | $Q_2$ | $|\psi\rangle$ | | H | | E | | H | | | | $|\psi\rangle$ |
| Ancilla | $Q_3$ | $|\downarrow\rangle$ | | H | | | | H | | | | |

**Fig. 1 | Three-qubit QEC and silicon-based three-qubit device. a**, Outline of the three-qubit phase-flip quantum error correcting code. The two-qubit CNOT gates entangle the three qubits, then the Hadamard (H) gates rotate the qubit basis for phase-flip errors. The decoding is the inverse of the encoding. Finally, the correction is performed by a three-qubit Toffoli gate. **b**, Scanning electron microscope image of the device. Scale bar, 100 nm. The screening gates (brown) are used to restrict the electric field of the plunger (green) and barrier (purple) gates. The three circles (red, green and blue) indicate the position of the triple-quantum-dot array. A further quantum dot shown as the grey circle is used as a charge sensor. The gates P1, P2, P3, B2 and B3 are connected to an arbitrary waveform generator to apply fast voltage pulses. The microwave control pulse for electric-dipole spin resonance is applied to the lower screening gate. **c**, Schematic cross section of the device. The line in the silicon quantum well shows the schematic triple-dot confinement potential. $J_{12}$ ($J_{23}$) represents the nearest-neighbour exchange coupling between $Q_1$ and $Q_2$ ($Q_2$ and $Q_3$).

microwave pulse displaces the quantum dot position, effectively creating an oscillating transverse magnetic field that induces electric-dipole spin resonance[27]. The two-qubit controlled phase (CZ) gate is implemented by adiabatically pulsing the exchange couplings $J_{12}$ and $J_{23}$ by the barrier gates B2 and B3, respectively (see Methods and Extended Data Fig. 3). To operate the qubit close to the charge-symmetry point, the effect of capacitive crosstalk between the plunger and barrier gates is suppressed by the virtual gate technique (see Methods). The spin qubits herein have an average $T_1$ relaxation time of 22 ms, inhomogeneous dephasing time $T_2^*$ of 1.8 μs and Hahn echo dephasing time $T_2^H$ of 43 μs (Extended Data Fig. 4). Because electron spins have orders of magnitude longer $T_1$ times compared with the dephasing times $T_2^*$ and $T_2^H$, we focus on the implementation of a phase-flip correction code in this work, whereas a bit-flip correction code can easily be assembled by introducing further single-qubit rotations.

First we demonstrate the ability to encode and decode the data qubit state. For simplicity, here we perform encoding of an input state on the equator of the Bloch sphere, $Q_2 = (|\downarrow\rangle + e^{i\phi}|\uparrow\rangle)/\sqrt{2}$ (Fig. 2a, $\phi$ is an azimuthal angle), which is encoded to a maximally entangled three-qubit Greenberger–Horne–Zeilinger (GHZ) state $|GHZ_\phi\rangle = (|\downarrow\downarrow\downarrow\rangle + e^{i\phi}|\uparrow\uparrow\uparrow\rangle)/\sqrt{2}$. The controlled NOT (CNOT) gates used in the encoding are decomposed to native CZ gates combined with the decoupling pulses to mitigate the local quasi-static phase noise. For the QEC implementation, a crucial property is that the encoded state is a genuine three-qubit GHZ-class state. We confirm this by characterizing the generated state using three-qubit quantum state tomography (Methods). In Fig. 2b (2c), the real part of the measured experimental density matrix $\rho$ for $\phi = 0$ ($\pi$) is plotted. We evaluate the state fidelities $F = \langle GHZ_\phi|\rho|GHZ_\phi\rangle$ for various $\phi$ (Fig. 2d) and confirm that all the states

**Fig. 2 | See next page for caption.**

**Fig. 2 | Encoding of three-qubit GHZ states and resonantly driven iToffoli gate. a**, Quantum circuit to generate three-qubit GHZ-class states. X (Y, Z) represents a π rotation about the $x$ ($y$, $z$) axis and X/2 (Y/2, Z/2) represents a π/2 rotation about the $x$ ($y$, $z$) axis. The two CNOT gates acting on the neighbouring qubits are implemented by the combination of single-qubit and two-qubit gates, as shown in the bottom half. The Y pulses in the middle of the sequence (surrounded by the purple box) are used to suppress the low-frequency single-qubit phase noise. **b,c**, Real parts of the measured density matrices of the three-qubit GHZ states ($\phi = 0$ in **b** and $\phi = \pi$ in **c**). **d**, Result of the GHZ state generation for various input states. The solid black line shows the average of

GHZ state fidelities, that is 0.866. The range above the threshold value 0.75 (0.5) to distinguish the GHZ-class states from the W-class (biseparable) states[40] is shown as the coloured band. **e**, Schematic energy diagram of the three-spin state. **f**, Resonance peaks of $Q_2$ for four different control qubit states at the exchange couplings $J_{12} = J_{23} = 4.5$ MHz. Here we define $\delta f = 0$ as the resonance condition when $Q_1Q_3 = |\downarrow\downarrow\rangle$. The circles show the measured $Q_2$ spin-up probabilities for the four different control qubit configurations. The solid lines show fitting with Gaussian functions. The traces are offset by 1 from each other for clarity. **g**, Schematic sequence of the measurement of the iToffoli gate truth table. **h**, Measurement result of the iToffoli gate truth table.

have fidelities above 0.75, the threshold to witness genuine GHZ-class states.

For correcting the decoded state, we implement a Toffoli-class three-qubit gate. The standard three-qubit Toffoli gate can be synthesized from 12 CNOT and two single-qubit gates[32,33] (excluding T gates that can be implemented in software), albeit that decoherence in our device does not allow this implementation with a reasonable fidelity. Alternatively, we use a single-step, resonantly driven iToffoli gate implemented by a resonant π pulse in the presence of simultaneous nearest-neighbour exchange couplings (Fig. 2e). Without the exchange couplings (left side of Fig. 2e), the four transitions associated with the $Q_2$ rotation are degenerate with a resonance frequency of $f_0$. The finite exchange couplings shift downward the energy levels of the antiparallel spin configurations. As a result, the resonance frequency of $Q_2$ is modulated as $f_0 + s_1 J_{12} + s_3 J_{23}$, in which $s_i = \pm 1/2$ is the spin number of $Q_i$. Under the condition of $J_{12} = J_{23}$ required for conditional phase synchronization (see Methods), a rotation of $Q_2$ with $Q_1Q_3 = |\downarrow\downarrow\rangle$ or $|\uparrow\uparrow\rangle$ corresponds to a controlled-controlled-rotation.

Figure 2f shows the spectra of $Q_2$ with four different ancilla qubit states $Q_1Q_3 = |\downarrow\downarrow\rangle$, $|\downarrow\uparrow\rangle$, $|\uparrow\downarrow\rangle$ and $|\uparrow\uparrow\rangle$ at $J_{12} = J_{23} = 4.5$ MHz, in which we observe the peak positions as expected from the exchange couplings. We use a resonant π pulse at $f_{MW} = f_1 (Q_1Q_3 = |\downarrow\downarrow\rangle)$ to implement our iToffoli gate, as this transition yields the highest visibility[34]. The iToffoli gate is a three-qubit gate equivalent to a Toffoli gate with an extra phase factor of $i$ on the ancilla qubits. To characterize its property, we prepare the eight possible three-spin eigenstates, apply the iToffoli gate and perform three-spin projective measurement (Fig. 2g,h). The readout errors are removed from the data based on the measured readout fidelities (see Methods). The Rabi frequency is chosen so that the off-resonant rotations for the $Q_1Q_3 = |\downarrow\uparrow\rangle/|\uparrow\downarrow\rangle$ subspaces are synchronized (see Methods). In Fig. 2h, as expected, the populations of $|\downarrow\downarrow\downarrow\rangle$ and $|\downarrow\uparrow\downarrow\rangle$ states are swapped, whereas the other states are essentially unaffected. From this result, we obtain a population transfer fidelity of our iToffoli gate as $\mathrm{Tr}(U_{\mathrm{expt}}U_{\mathrm{ideal}})/8 = 0.96$, in which $U_{\mathrm{expt}}$ ($U_{\mathrm{ideal}}$) represents the experimental (ideal) gate action on the eigenstates (see Methods and Extended Data Fig. 5e–g for the result of the full quantum process tomography). In addition, we perform a calibration of the pulse duration and timing to eliminate unwanted phase accumulation on $Q_2$ (see Methods). Note that the dephasing and phase accumulation on the ancilla qubits do not affect the error correction outcome.

We then turn to the implementation of the phase-flip correcting code. Figure 3a shows the quantum circuit diagram. The three-qubit operation U serves to encode the data qubit state $|\psi\rangle$ to the three-qubit entangled state. The exact implementation of U is shown in the bottom half of the figure, and it is equivalent to the two CNOT gates shown in Fig. 2a, except for the single-qubit gates that do not affect the function of the QEC. Here the data qubit state $|\psi\rangle = \alpha|\downarrow\rangle + \beta|\uparrow\rangle$ is encoded to a phase-sensitive three-qubit state $\alpha|{+}{+}{+}\rangle + \beta|{-}{-}{-}\rangle$, in which $|\pm\rangle = (|\downarrow\rangle \pm |\uparrow\rangle)/\sqrt{2}$ are the eigenstates of the Pauli X operator. For a phase-flip error with a flip rate of $p$ on $Q_2$, the decoded state is $\sqrt{1-p}|\downarrow\rangle(\alpha|\downarrow\rangle + \beta|\uparrow\rangle)|\downarrow\rangle + \sqrt{p}|\uparrow\rangle(\beta|\downarrow\rangle + \alpha|\uparrow\rangle)|\uparrow\rangle$ (see Extended Data Table 1 for the cases with an error on ancilla). The correcting procedure

is implemented so that $Q_2$ is flipped only when $Q_1Q_3 = |\uparrow\uparrow\rangle$ by applying π pulses on the ancilla qubits followed by the iToffoli gate, resulting in a product state of $Q_2 = \alpha|\downarrow\rangle + \beta|\uparrow\rangle$ and $Q_1Q_3 = \sqrt{1-p}|\uparrow\uparrow\rangle + i\sqrt{p}|\downarrow\downarrow\rangle$. Now the data qubit state is the same as the input state regardless of $p$. This is demonstrated in Fig. 3b, in which we estimate the process fidelity of the data qubit for various intentional one-qubit errors (see Methods for details of the quantum process tomography). We implement the one-qubit error as a phase rotation with a known rotation angle $\theta$, which is equivalent to a phase-flip error with $p = \sin^2(\theta/2)$. Therefore, without the correction, the process fidelity oscillates as a function of $\theta$, shown as the black points. With the correction, the oscillation vanishes, and it confirms the basic function of the phase-flip correcting code (corrected $Q_i$ error means that a phase-flip error is applied to only $Q_i$ and the correction is performed). When there is no error ($\theta = 0$), the process fidelity slightly decreases after the correction. This can be attributed to the infidelity of the iToffoli gate projected to the data qubit subspace. Furthermore, we show that the state of ancilla

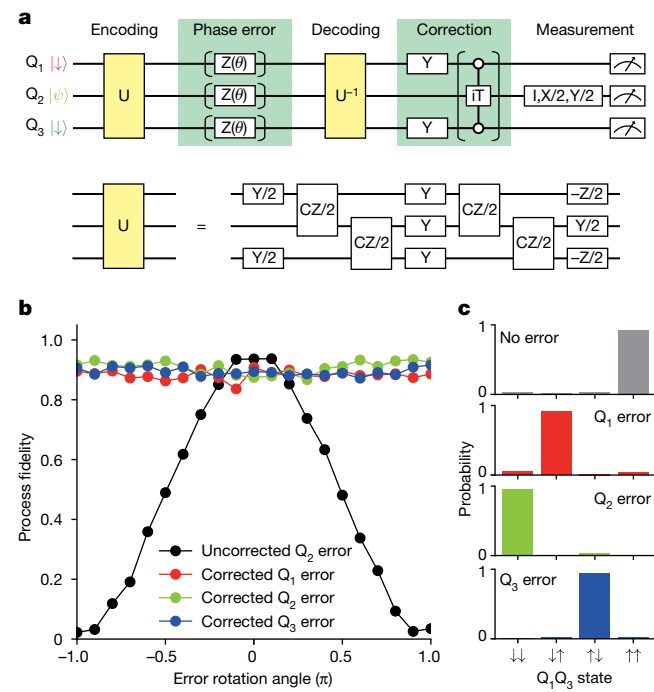

**Fig. 3 | One-qubit phase error correction. a**, Schematic of the quantum circuit. The operation U used for encoding and decoding is decomposed into the single-qubit and two-qubit gates, as shown in the lower half of the figure. **b**, Result of one-qubit phase error correction. In the case of uncorrected, we omit the iToffoli gate and the rest of the quantum circuit is the same as the one for the corrected case. For the ideal case without gate infidelities, the uncorrected fidelity oscillates from 0 to 1 and the corrected fidelities are always equal to 1. **c**, Ancilla qubit measurement results. Note that, owing to the implementation of our correcting procedure, the resulting population of the ancilla qubit state is flipped as compared with the implementation using a standard Toffoli gate[11].

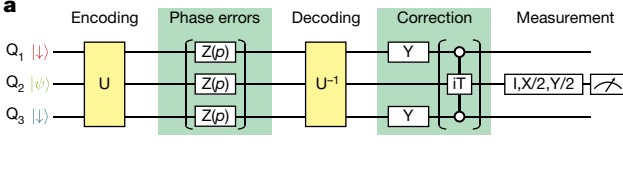

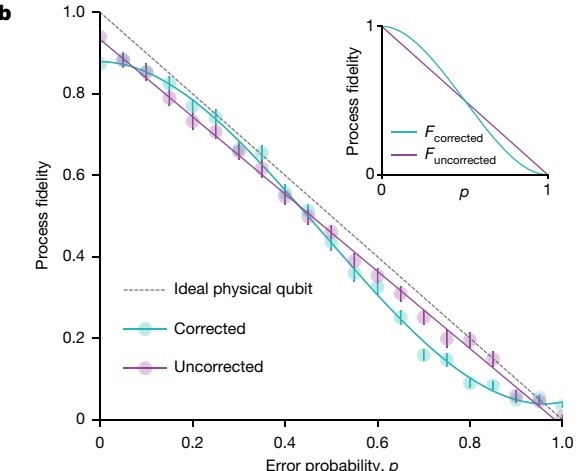

**Fig. 4 | Three-qubit phase error correction. a**, Schematic of the quantum circuit for three-qubit phase error correction. The phase error Z($p$) is a virtual phase rotation with a rotation angle of 2arcsin($\sqrt{p}$), which results in an effective error rate of $p$. We prepare the data qubit input state $|\psi\rangle$ by initialization to a spin-down state and a subsequent single-qubit rotation I, X/2, Y/2 or X. **b**, Measured process fidelities for the corrected and uncorrected cases. Each data point is obtained by averaging 2,000 experiments that are segmented into 1,000 experiments with interleaved qubit frequency calibrations. The error bars are obtained by a Monte Carlo resampling method[41] and represent 1$\sigma$ from the mean. Inset, calculated fidelities for the ideal cases without gate errors. **c**, Schematic of the quantum circuit for three-qubit dephasing error correction. The waiting time $t_w$ is the time interval between the last single-qubit rotation in U and the first single-qubit rotation in U$^{-1}$. The

deviation of the purple curve from the black curve reflects the gate infidelities in the encoding and decoding. **d**, Comparison of the state fidelities of the corrected and uncorrected qubits. In the case of the physical qubit, we perform a Ramsey measurement with varying waiting time $t_w$ between the first π/2 pulse and the pre-rotation for tomographic readout. Each data point is obtained by averaging 3,000 experiments that are segmented into 1,000 experiments with interleaved qubit frequency calibrations. The data acquisition time is the same for all traces in this figure. The solid curves are guides to the eye obtained by fitting to a general exponential decaying function[42] $F(t) = (1 + \alpha \exp(-(t/T_2)^n))/2$, with $\alpha = 0.492 \pm 0.005$, $0.432 \pm 0.008$ and $0.464 \pm 0.007$, $T_2 = 1.44 \pm 0.02$, $1.36 \pm 0.04$ and $1.12 \pm 0.03$ μs, and $n = 1.90 \pm 0.08$, $2.1 \pm 0.2$ and $1.68 \pm 0.08$ for the physical qubit, corrected qubit and uncorrected qubit, respectively. The errors are 1$\sigma$ from the mean.

qubits reflects the error on the encoded qubit state (error detection). We measure the joint probability of the ancilla qubits $Q_1$ and $Q_3$ for the four possible cases with no error or a full π flip error. We observe that the measured ancilla states correctly reflect the error occurring on the encoded three-qubit state (Fig. 3c).

Errors in actual quantum computers probably occur on all qubits simultaneously rather than on only one of the qubits. We verify the performance of our error correcting code in such a case in which all errors have the same effective error rate of $p$ as per the common assumption in QEC[14] (Fig. 4a). Without the correction, the data qubit process fidelity linearly decreases as $p$ is increased. When the error correction is applied, errors on two and three qubits remain uncorrected, resulting in a process fidelity insensitive to $p$ up to the first order, $F(p) = 1 - 3p^2 + 2p^3$ (ref. [14]) (see Fig. 4b inset). The quadratic dependence on $p$ is a crucial property of QEC and it ideally results in an improvement of the fidelity for $p < 0.5$. We confirm this crucial property in Fig. 4b, in which the measured process fidelity with the correction is plotted as the cyan curve. A quadratic function fits well to the data (see Extended Data Fig. 6 for a comparison between different fitting models). If we allow the first-order term in the fit, it is $0.0 \pm 0.1$ (the error is 1$\sigma$), representing a marked reduction of the first-order sensitivity as compared with the uncorrected case. As for the fidelity enhancement, the corrected qubit shows improvements in the range $p < 0.429 \pm 0.003$ (the threshold is obtained by comparing the two fitted curves in Fig. 4b, the error is 1$\sigma$). Although the corrected fidelities are always lower than those of the ideal uncorrected qubit in the present experiment (dashed grey line in

Fig. 4b), improvement of the coherence times and thereby the fidelity of the iToffoli gate, which primarily limits the performance in the corrected case, would ameliorate the situation. In silicon spin qubits, the intrinsic phase error is more like a quasi-static phase shift rather than a sudden phase flip. In our device, the phase shift is mainly caused by the fluctuating spins of surrounding $^{29}$Si nuclei. To demonstrate the effectiveness of our error correcting code to this type of phase error, we measure the dephasing of the encoded three-qubit state (Fig. 4c,d). As predicted from the ability to correct small phase errors in Fig. 4b, the initial slope of the fidelity decay is suppressed as compared with that of an uncorrected encoded qubit. Overall, these results show a successful implementation of three-qubit phase-correcting code in silicon.

In conclusion, we have demonstrated the generation of the various three-qubit entangled states, the effective single-step resonantly driven iToffoli gate and the fundamental properties of three-qubit QEC in silicon. Extending the experiment to a larger scale would require a more flexible feedback-based correcting rotation. This would be limited by the slow spin measurement and initialization by energy-selective tunnelling, which also pose a challenge to complete the error correction (or detection) before the phase coherence is completely lost. Substantial improvements should be possible by switching to the singlet-triplet readout, in which high-fidelity spin measurements in a few μs (refs. [35,36]), orders of magnitude shorter than the phase coherence time with dynamical decoupling[21], are routinely achieved. Along with the recent advances in scalable device design[37], electronics[38] and gate fidelities[6–8], we anticipate that it will become possible to demonstrate

more sophisticated quantum error correcting codes in a large-scale silicon-based quantum processor.

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

# Methods

## Quantum dot device

The triple-quantum-dot device is identical to the one characterized in ref. [24]. The device is fabricated using an isotopically natural, undoped Si/SiGe heterostructure. The ohmic contacts are made by phosphorus ion implantation. Standard electron-beam lithography and lift-off techniques are used to fabricate the overlapping aluminium gates and the micro-magnet.

## Experimental setup

The GHZ state tomography and the iToffoli gate characterization (Fig. 2) are performed using the experimental setup as described in ref. [24]. In what follows, we detail the modified experimental setup used for the QEC experiments in Figs. 3 and 4. The sample is cooled down in a dry dilution refrigerator (Oxford Instruments Triton 300) to a base electron temperature of around 40 mK. The configuration of d.c. lines is the same as in the previous report[24]. Control pulses are generated by four Keysight M3201A arbitrary waveform generator modules in a Keysight M9019A PXIe chassis (16 channels running at 500 MSa s$^{-1}$). The plunger (P1, P2 and P3), barrier (B2 and B3) and sensor plunger gates are connected to the outputs of the arbitrary waveform generator, each of which is filtered by a Mini Circuits SBLP-39+ Bessel low-pass filter. The filtering results in a minimum pulse rise/fall time of approximately 15 ns. Microwave signals are generated by three vector microwave signal generators (two Keysight E8267D and a Rohde & Schwarz SGS100A with an SGU100A upconverter). Each microwave signal is single sideband I/Q modulated to prevent unintentional spin rotations owing to microwave carrier leakage. Furthermore, we use pulse modulation to further suppress the bleed-through signal during the initialization and readout stages. The outputs of the three signal generators are combined at room temperature and connected to the lower screening gate. Radio-frequency reflectometry is used for fast measurement of the charge sensor conductance. The right reservoir of the charge sensor quantum dot in Fig. 1b is connected to a tank circuit with an inductance of 1.2 μH and a resonance frequency of 181 MHz. The reflected signal is amplified and demodulated, then digitized using an AlazarTech ATS9440 digitizer card.

## Three-spin initialization and measurement

The three-spin initialization and measurement are performed as follows. The numbers $(n_1n_2n_3)$ indicate the respective number of electrons in the left, centre and right quantum dots. We collect 400 to 3,000 single-shot outcomes to obtain the measured probabilities. The labels A–E represent the gate voltage configurations depicted in Extended Data Fig. 1c.

1. Unload electrons in the left and centre quantum dots by biasing gate voltages so that the ground state charge configuration is (001) (A). The duration is 100 μs.
2. Initialize $Q_1$ by means of spin-selective tunnelling by biasing the voltages so that the charge configuration is near the (101)–(001) boundary (B). The duration is 750 μs.
3. Shuttle the electron in the left quantum dot to the centre quantum dot by biasing the voltages so that the ground charge configuration is (011) (C). No intentional gate voltage ramp is used. The typical pulse rise time is 15 ns owing to the low-pass filter. We wait for 1 μs in the (011) configuration.
4. Initialize $Q_1$ by means of spin-selective tunnelling by biasing the voltages so that the charge configuration is near the (011)–(111) boundary (D). The duration is 750 μs.
5. Initialize $Q_3$ by means of spin-selective tunnelling by biasing the voltages so that the charge configuration is near the (110)–(111) boundary (E). The duration is 750 μs.
6. Qubit manipulation in the (111) configuration (F). The typical duration is 5 μs. There is an extra waiting time of 50 μs to reduce the effect of heating by the microwave pulses.

7. Read out $Q_1$ by means of spin-selective tunnelling by biasing the voltages so that the charge configuration is near the (011)–(111) boundary (D). The total duration is 600 μs. The data for readout is collected for the first 200 μs. The extra waiting time of 400-μs duration facilitates the initialization of $Q_1$.
8. Perform controlled rotation between $Q_1$ and $Q_2$ to project the $Q_2$ state to $Q_1$ in (111). Here we pulse the virtual B2 gate to turn on $J_{12}$ at the charge-symmetry point. Because $Q_1$ is initialized to a spin-down state during the previous readout stage, for a $Q_2$ input state $\alpha|\uparrow\rangle + \beta|\downarrow\rangle$, the resulting $Q_1Q_2$ state is $\alpha|\uparrow\downarrow\rangle + e^{i\theta}\beta|\downarrow\uparrow\rangle$, in which $e^{i\theta}$ is a phase factor that does not affect the readout. The duration is 1 μs. There is an extra waiting time of 50 μs to reduce the effect of heating by the microwave pulse.
9. Read out $Q_2$ by means of spin-selective tunnelling of $Q_1$ by biasing the voltages so that the charge configuration is near the (011)–(111) boundary (D). The duration is 200 μs.
10. Read out $Q_3$ by means of spin-selective tunnelling by biasing the voltages so that the charge configuration is near the (110)–(111) boundary (E). The duration is 500 μs.

## Virtual gate

The effect of capacitive couplings between the gates is suppressed by the virtual gate technique. We measure the capacitive couplings between the gates and construct the virtual gate as follows. The cross-talk between the exchange couplings is not taken into account. The virtual gate voltages vB2 and vB3 are used to control the exchange couplings.

$$\begin{pmatrix} vP1 \\ vB2 \\ vP2 \\ vB3 \\ vP3 \end{pmatrix} = \begin{pmatrix} 1 & 0.30 & 0.54 & 0.14 & 0.17 \\ 0 & 1 & 0 & 0 & 0 \\ 0.61 & 0.35 & 1 & 0.25 & 0.31 \\ 0 & 0 & 0 & 1 & 0 \\ 0.15 & 0.10 & 0.46 & 0.31 & 1 \end{pmatrix} \begin{pmatrix} \Delta P1 \\ \Delta B2 \\ \Delta P2 \\ \Delta B3 \\ \Delta P3 \end{pmatrix}.$$

## Single-qubit and two-qubit gates

The single-qubit rotations about the $x$ and $y$ axes are performed by applying microwave voltage pulses resonant with the Zeeman splitting of each spin qubit. The microwave voltage results in an effective out-of-plane a.c. magnetic field by the micro-magnet, which induces electric-dipole spin resonance. The spin qubits have typical resonance frequencies of 19,942.6 MHz ($Q_1$), 20,372.6 MHz ($Q_2$) and 20,923.2 MHz ($Q_3$). We use a shaped raised-cosine pulse with a duration of 124 (62) ns to implement a single-qubit π (π/2) pulse. For the spectroscopy measurements in Fig. 2f, we use a Gaussian pulse (truncated at ±2$\sigma$). The phase rotation is virtually implemented by shifting the reference phase of the I/Q modulation waveform. Wherever possible, the single-qubit gates are applied in parallel. The two-qubit CZ gate is implemented by adiabatically pulsing the exchange coupling by the barrier gates. To guarantee the adiabaticity, we use a shaped cosine pulse[6] with a duration of 50 ns to implement the CZ/2 gates, which results in a nominal peak exchange coupling of 10 MHz. During the experiments in the main text, the coupling strengths are fine-tuned to account for the conditional phase accumulation owing to the residual couplings of about 0.2 MHz (Extended Data Fig. 3d–f). We set the minimum interval between pulses to 20 ns to avoid the pulse interference owing to reflection.

## Three-qubit iToffoli gate

The resonantly driven iToffoli gate consists of the three stages in Extended Data Fig. 5a. In the main text, the population transfer property of the iToffoli gate is shown. For that, we set $f_{Rabi} = J/\sqrt{3}$ ($J = J_{12} = J_{23}$) so that the off-resonant rotation in the $Q_1Q_3 = |\uparrow\downarrow\rangle/|\downarrow\uparrow\rangle$ subspaces is a 2π rotation. Furthermore, to obtain a correct quantum action, any unwanted phase accumulations on the three-qubit state have to

be calibrated out. This can be achieved by setting an appropriate exchange pulse duration of $t_{tot} = t_{dc1} + t_{MW} + t_{dc2}$ and pulse timing of $\delta t = t_{dc1} - t_{dc2}$ (ref. [33]). In theory, by setting the optimal exchange pulse duration to $t_{tot} = \pi(4 + \sqrt{3} - \sqrt{13})/J$, the conditional phases between the $Q_1Q_3 = |\uparrow\downarrow\rangle, |\downarrow\uparrow\rangle$ and $|\uparrow\uparrow\rangle$ subspaces can be eliminated[32]. For an exchange coupling of 4.5 MHz, it is 473 ns. In the experiment, we typically use a 460-ns-long rectangular pulse, which is shorter than the theoretical length owing to the finite pulse bandwidth. The microwave pulse timing $\delta t$ is then adjusted to eliminate the conditional phase between the $Q_1Q_3 = |\downarrow\downarrow\rangle$ and the other subspaces. For the subspaces in which $Q_2$ spin flip does not occur, shifting $\delta t$ does not affect the outcome. In the case in which $Q_2$ flips, when $\delta t = 0$, (quasi-)static phase accumulation is fully cancelled out by the spin-echo effect. The conditional phase in this case can be adjusted by varying $\delta t$ because for finite $\delta t$, the echo works only partially and there is a phase accumulation of $2\pi(f_1 - f_0)\delta t$. The remaining single-qubit phase offset is removed by a virtual single-qubit phase rotation. The phase offsets on the ancilla qubits are uncalibrated in the QEC experiments, although they can be calibrated out similarly. In Extended Data Fig. 5b, we illustrate the experimental sequence to calibrate the iToffoli gate phase accumulation. Extended Data Fig. 5c shows an example of an uncalibrated iToffoli gate and Extended Data Fig. 5d shows a phase measurement after the calibration. In the QEC experiments, this calibration is performed just before the data acquisition to minimize the influence of the slow drift of the resonance frequencies.

## Readout error removal

For each of the experiments in which the readout errors are removed, we perform a reference measurement to obtain the readout fidelities. The spin-down (spin-up) readout fidelity $F_{\downarrow i}$ ($F_{\uparrow i}$) is directly obtained by preparing a spin-down (spin-up) state and a projective measurement of $Q_i$. Using the measured readout fidelities, we correct the raw probabilities $\mathbf{P}_M = (P_{\downarrow\downarrow\downarrow}, ..., P_{\uparrow\uparrow\uparrow})$ as $\mathbf{P} = (F_1 \otimes F_2 \otimes F_3)^{-1}\mathbf{P}_M$, in which $F_i = \begin{pmatrix} F_{\downarrow i} & 1 - F_{\uparrow i} \\ 1 - F_{\downarrow i} & F_{\uparrow i} \end{pmatrix}$ and $\mathbf{P}$ is the corrected probabilities used for maximum likelihood estimation.

## Measurement of the iToffoli gate truth table

To constrain all the elements of the truth table to be non-negative, we use a maximum likelihood procedure as follows. The input is a set of 64 measured probabilities $P_{ij}$, in which the input is the $i$th eigenstate and the measurement is projected at the $j$th eigenstate. The readout errors are removed following the procedure above. We then minimize a cost function $C(P_{11}^{MLE}, ..., P_{88}^{MLE}) = \sum_{i,j=1}^{8}(P_{ij}^{MLE} - P_{ij})^2$ for non-negative parameters $P_{ij}^{MLE}$. We constrain $P_{ij}^{MLE}$ so that the sum of probabilities in one cycle of data acquisition is unity, that is, $\sum_{j=1}^{8} P_{ij}^{MLE} = 1$.

## Quantum state tomography

Owing to the noise in the experiment, the density matrix obtained by a linear inversion is not always physical. Therefore, we use a maximum likelihood estimation to restrict the density matrix to be physical. We start from a Cholesky decomposition of a physical density matrix $\rho = T^{\dagger}T/\mathrm{Tr}(T^{\dagger}T)$, in which $T$ is a complex lower triangular matrix with real diagonal elements. $T$ has $2^{2D}$ ($D$ is the number of qubits; $D = 1$ in Fig. 4d and $D = 3$ in Fig. 2d) real parameters $\mathbf{t} = (t_1, ..., t_{2^{2D}})$ and we minimize the cost function

$$C(\mathbf{t}) = \sum_{v=1}^{2^{2D}} \frac{(\langle \psi_v | \rho(\mathbf{t}) | \psi_v \rangle - P_v)^2}{2\langle \psi_v | \rho(\mathbf{t}) | \psi_v \rangle},$$

in which $P_v$ is the measured probability projected at a basis $|\psi_v\rangle$. To determine the $2^{2D}$ parameters, the projection outcomes for linearly independent pre-rotations $(I, X/2, Y/2, X)^{\otimes D}$ are used. To remove the error that could be introduced by the X pre-rotation, the projection outcomes for the X pre-rotations are calculated from the corresponding I rotation outcomes[39].

## Quantum process tomography

We perform quantum process tomography to verify the process matrix and fidelity (Figs. 3 and 4 and Extended Data Fig. 5). The input state $|\psi\rangle$ is prepared by a spin-down initialization followed by a single-qubit rotation $R_i \in (I, X/2, Y/2, X)^{\otimes D}$ ($D$ is the number of qubits; $D = 1$ in Fig. 4b and $D = 3$ in Extended Data Fig. 5e). After the quantum operations, tomographic readout of the resulting state is performed similarly to the case of quantum state tomography. For a quantum operation $E$ acting on an input density matrix $\rho_{in}^k$, the density matrix of the output state can be written as follows,

$$E(\rho_{in}^k) = \sum_{m,n=1}^{2^{2D}} B_m \rho_{in}^k B_n^{\dagger} \chi_{mn}, \qquad (1)$$

in which $\chi$ is the process matrix defined with respect to the Pauli operators $B = (I, \sigma_x, \sigma_y, \sigma_z)^{\otimes D}$. Linear inversion of equation (1) can be performed to obtain a process matrix. However, the process matrix obtained in this way does not necessarily satisfy the physical conditions owing to the noise in the experiment. As in the state tomography, we can obtain an estimate of the physical process matrix by a maximum likelihood estimation. We start from a Cholesky decomposition $\chi = S^{\dagger}S/\mathrm{Tr}(S^{\dagger}S)$, in which $S$ is a lower triangular matrix with real diagonal elements. $S$ is parametrized by $2^{4D}$ real parameters $\mathbf{s} = (s_1, ..., s_{2^{4D}})$ and we use a cost function $L(\mathbf{s})$ as follows,

$$L(\mathbf{s}) = \sum_{k,l=1}^{2^{2D}} [P^{kl} - \sum_{m,n=1}^{2^{2D}} \chi_{mn} \mathrm{Tr}(M_l B_m \rho_{in}^k B_n^{\dagger})]^2, \qquad (2)$$

in which $P^{kl}$ is the measured probability projected at $|\downarrow\rangle$ ($D = 1$) or $|\downarrow\downarrow\downarrow\rangle$ ($D = 3$) when an input state $\rho_{in}^k$ is prepared and an observable $M_l$ is measured. We numerically minimize the cost function to obtain the most probable estimate of physical $\chi$. Then the process fidelity is calculated as $\mathrm{Tr}(\chi_{ideal}\chi)$, in which $\chi_{ideal}$ is an ideal process matrix.

## Data availability

The data that support findings in this study are available from the Zenodo repository at https://doi.org/10.5281/zenodo.6601051.

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

**Acknowledgements** This work was supported financially by Core Research for Evolutional Science and Technology (CREST), Japan Science and Technology Agency (JST) (JPMJCR15N2 and JPMJCR1675), JST Moonshot R&D grant no. JPMJMS2065, MEXT Quantum Leap Flagship Program (MEXT Q-LEAP) grant no. JPMXS0118069228 and JSPS KAKENHI grant nos. 18H01819, 19K14640 and 20H00237. T.N. acknowledges support from a Murata Science Foundation Research grant and JST PRESTO grant no. JPMJPR2017.

**Author contributions** K.T. and A.N. fabricated the device and performed the measurements. T.N. and T.K. contributed the data acquisition and discussed the results. K.T. wrote the paper with input from all co-authors. S.T. supervised the project.

**Competing interests** The authors declare that they have no competing interests.

**Additional information**
**Correspondence and requests for materials** should be addressed to Kenta Takeda or Seigo Tarucha.

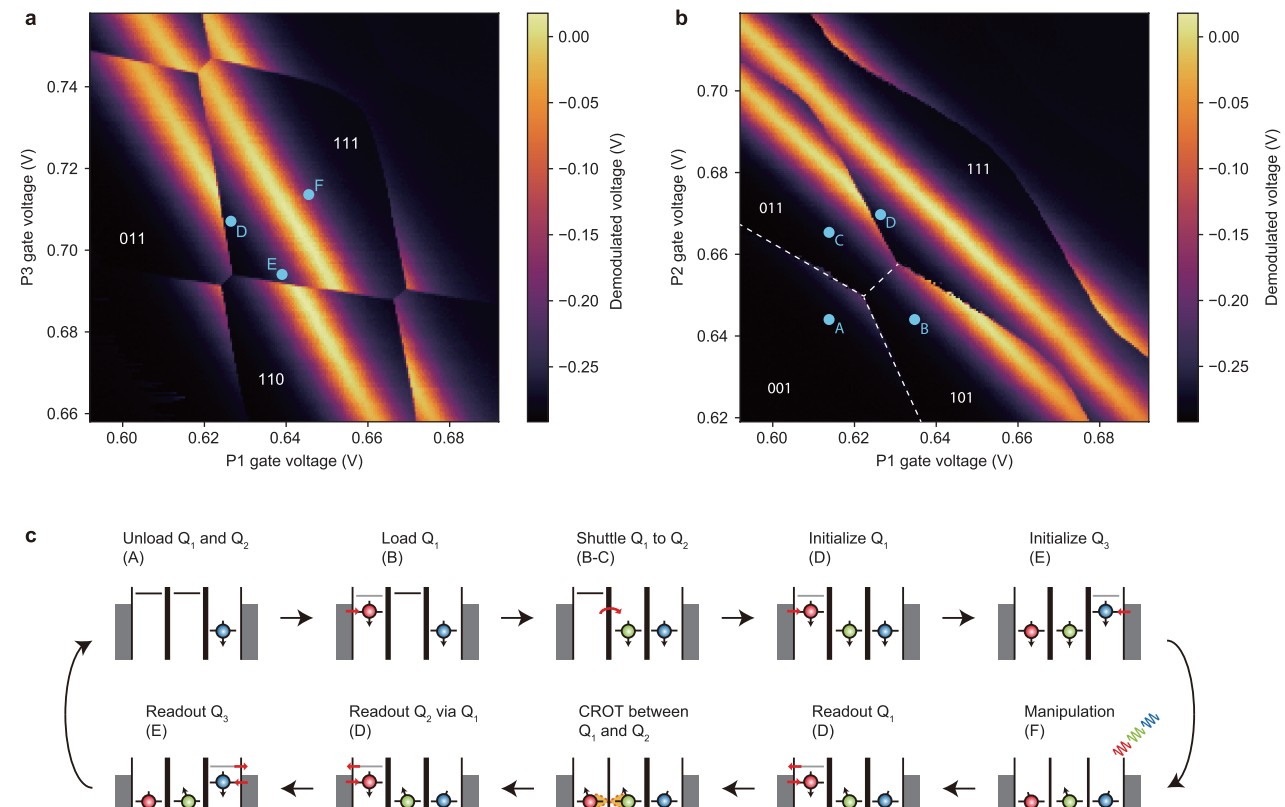

**Extended Data Fig. 1 | Three-spin initialization and measurement.** The numbers ($n_1 n_2 n_3$) represent the respective electron occupations in the right, centre and left quantum dots. The light blue circles with labels A–F show the initialization, readout and manipulation bias configurations. **a**, Charge stability diagram measured as a function of the P1 and P3 gate voltages. The variation of the background signal is due to the Coulomb oscillation of the sensor quantum dot. **b**, Charge stability diagram measured as a function of the P1 and P2 gate voltages. The dashed white lines are guides to the eye for the position of faint charge transition lines, which could be visible by retuning of the sensor quantum dot. **c**, Schematic of the three-spin initialization and measurement.

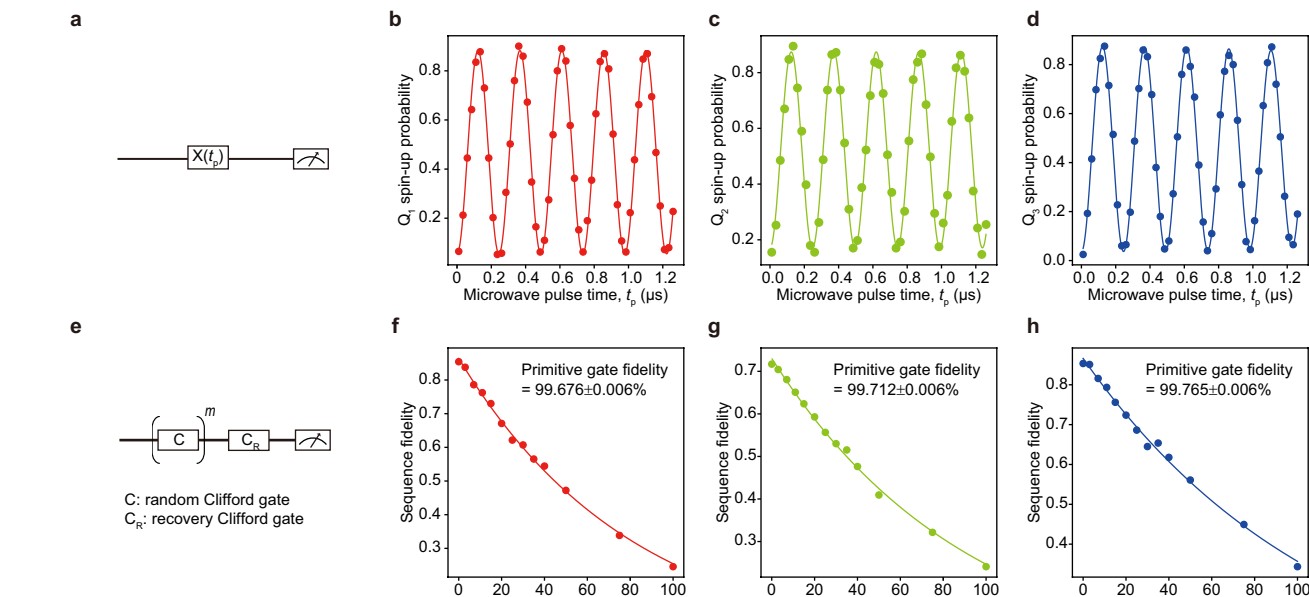

**Extended Data Fig. 2 | Single-qubit rotations.** All measurements are performed with all qubits initialized to spin-down and the exchange couplings turned off. **a**, Rabi oscillation measurement sequence. $t_p$ is the duration of the microwave pulse. **b**–**d**, Rabi oscillation measurement results. The microwave amplitude is adjusted so that the Rabi frequency is 4 MHz. **e**, Schematic sequence of the randomized benchmarking measurement. We prepare 16 randomly generated Clifford gate sequences and average the outcomes to obtain the sequence fidelities. **f**–**h**, Randomized benchmarking results. The implementation is the same as in, for example, refs. [7,24]. We perform two sets of benchmarking measurements, one designed to obtain an ideal spin-up outcome and the other designed to obtain an ideal spin-down outcome, wherein both cases the measurement is projected at a spin-up state. The sequence fidelity $F(m)$ is then defined as $F(m) = F_\uparrow(m) - F_\downarrow(m)$, in which $F_\uparrow(m)$ ($F_\downarrow(m)$) is the measured sequence fidelity for the spin-up (spin-down) final state. Each dataset is fit by an exponential decay $F(m) = Vp^m$ to extract the depolarizing parameter $p$ and visibility $V$. The primitive gate fidelity shown in each figure is obtained as $1 - (1 - p)/(2 \times 1.875)$, in which the factor 1.875 is the average number of primitive gates per Clifford gate. The errors are $1\sigma$ from the mean.

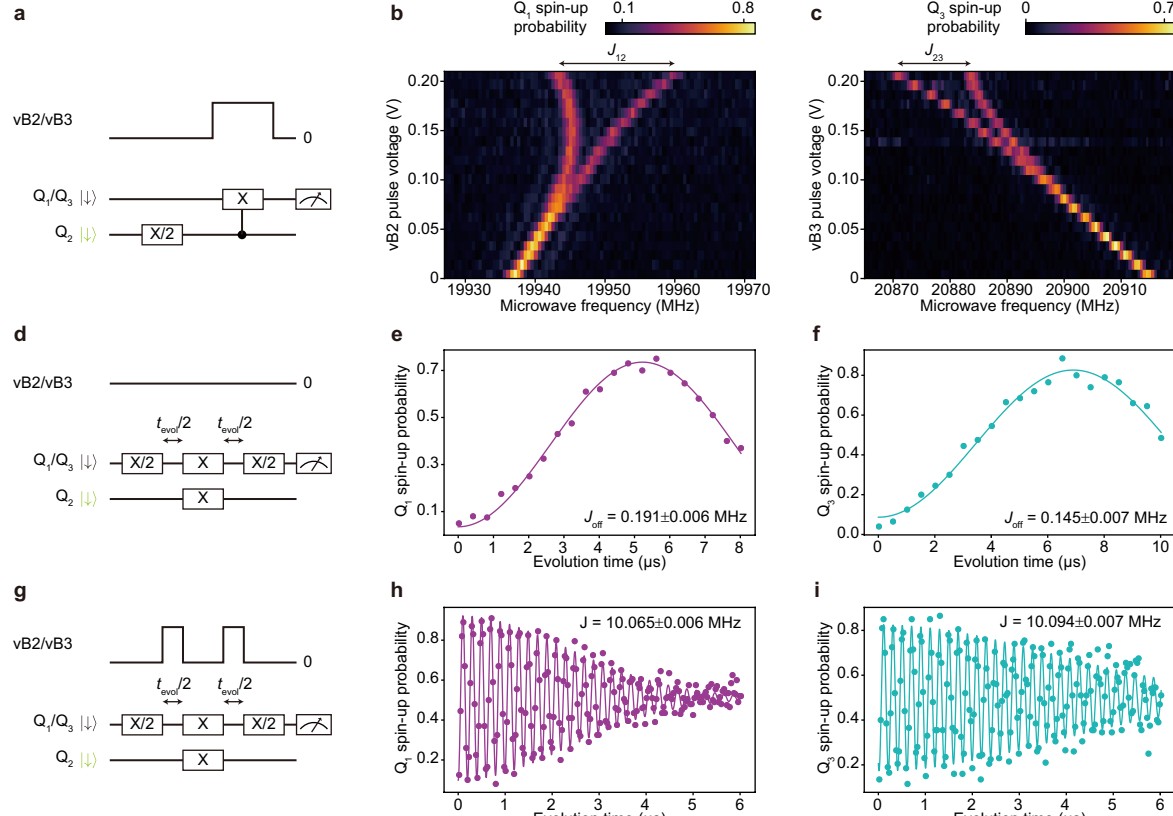

**Extended Data Fig. 3 | Two-qubit couplings.** All measurements are performed with all qubits initialized to spin-down. **a**, Schematic sequence of the exchange spectroscopy measurement. To narrow the resonance peaks, the microwave power for the controlled rotation is decreased by 12 dB from the values used for single-qubit rotations. vB$i$ ($i$ = 2, 3) represents a virtual barrier gate voltage. **b**,**c**, Results of the exchange spectroscopy measurements. In each figure, the separation of the two peaks corresponds to the exchange coupling. The background slope of the resonance frequency is due to the displacement of the quantum dot position in the micro-magnet field gradient. The frequency offset from the values in Methods is due to the decay of the persistent current in the superconducting magnet. **d**, Schematic sequence of the residual exchange-coupling measurement. **e**,**f**, Results of the measurement of residual exchange couplings between neighbouring qubits. Each dataset is fit with a sinusoidal function $P(t_{evol}) = V\sin(\pi t_{evol} J_{off})$ to extract the residual exchange coupling $J_{off}$. $V$ is the visibility of the oscillation. The errors are 1$\sigma$ from the mean. **g**, Schematic sequence of the decoupled CZ oscillation measurement. **h**,**i** Typical decoupled CZ oscillations. The solid lines show the fit to a Gaussian decay. The decay times are 3.27 ± 0.08 μs (**h**) and 5.2 ± 0.3 μs (**i**). Here we adjust the virtual barrier gate voltages so that the exchange coupling is roughly 10 MHz. All errors are 1$\sigma$ from the mean.

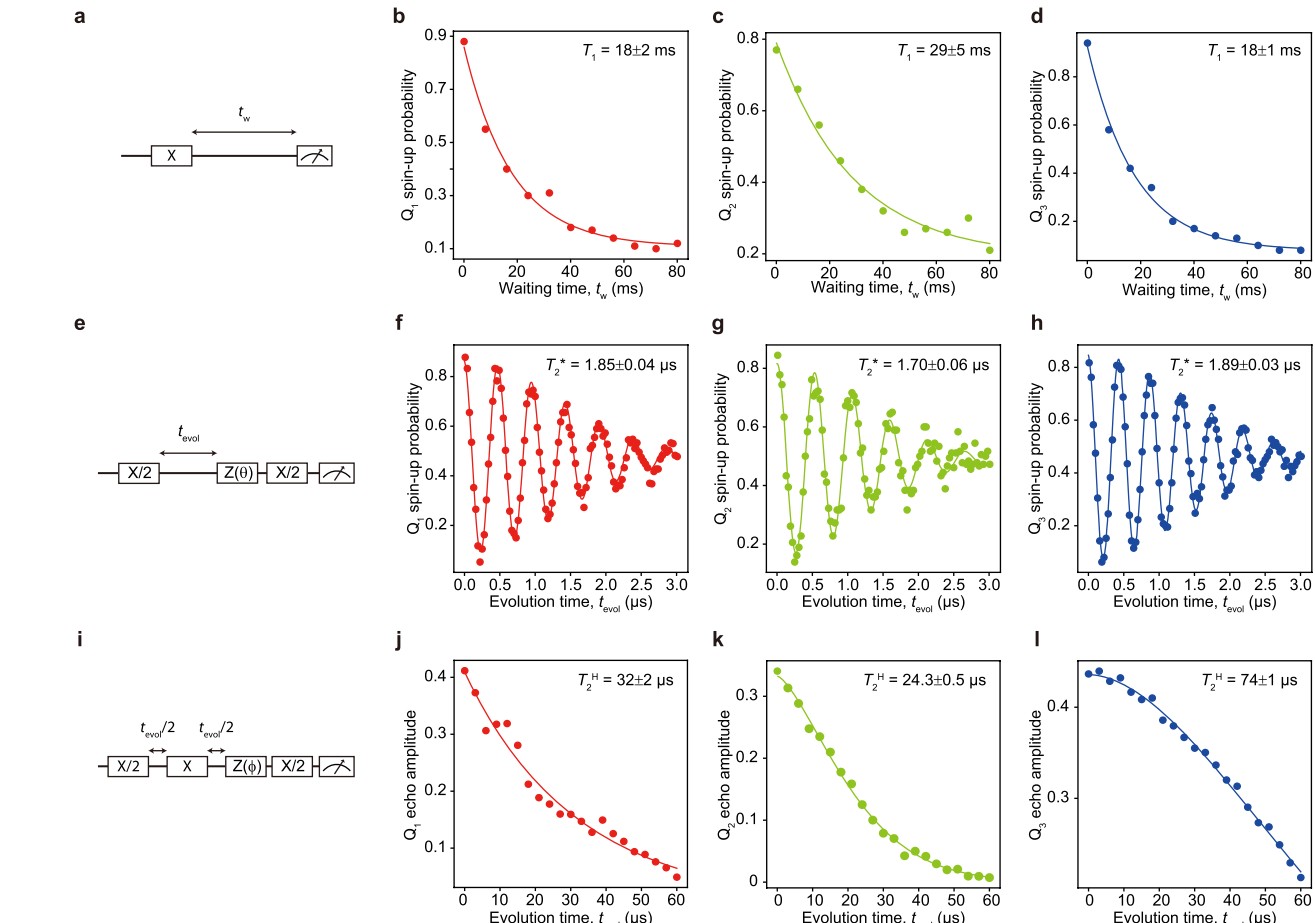

**Extended Data Fig. 4 | Coherence times.** All measurements are performed with all qubits initialized to spin-down and the exchange couplings turned off. All errors are 1$\sigma$ from the mean. **a**, Schematic sequence of the $T_1$ measurement. The qubit state is measured after the preparation of a spin-up excited state and an idle time of $t_w$. **b**–**d**, $T_1$ measurement results. Each dataset is fit by an exponential decay to extract the $T_1$ relaxation time. **e**, Schematic sequence of the Ramsey interferometry. Instead of detuning the microwave frequency, we vary the phase of the second microwave pulse as $\theta = 2\pi t_{evol} \times (2\,\text{MHz})$ such that we observe an oscillation at about 2 MHz. **f**–**h**, Ramsey interferometry measurement results. To extract the $T_2^*$ inhomogeneous dephasing time, each dataset is fitted with a Gaussian decay function $P(t_{evol}) = A\exp\left(-\left(\frac{t_{evol}}{T_2^*}\right)^2\right)\cos(2\pi(\delta f)t_{evol} + \phi) + B$,

in which $A$ and $B$ are the constants to account for the readout fidelities, $\delta f$ is the oscillation frequency and $\phi$ is the phase offset. The integration time is about 70 s for all traces. The larger scattering of the data points for $Q_2$ (**g**) is due to the longer pulse cycle and less averaging. **i**, Schematic sequence of the Hahn echo measurement. **j**–**l**, Hahn echo results. For each dataset, the echo time $T_2^H$ is extracted by fitting with an exponential decay function $P(t_{evol}) = V\exp\left(-\left(\frac{t_{evol}}{T_2^H}\right)^\gamma\right)$, in which $V$ is the visibility and $\gamma$ is the exponent. The exponents are $\gamma = 0.98 \pm 0.09$ ($Q_1$), $1.46 \pm 0.05$ ($Q_2$) and $1.83 \pm 0.07$ ($Q_3$).

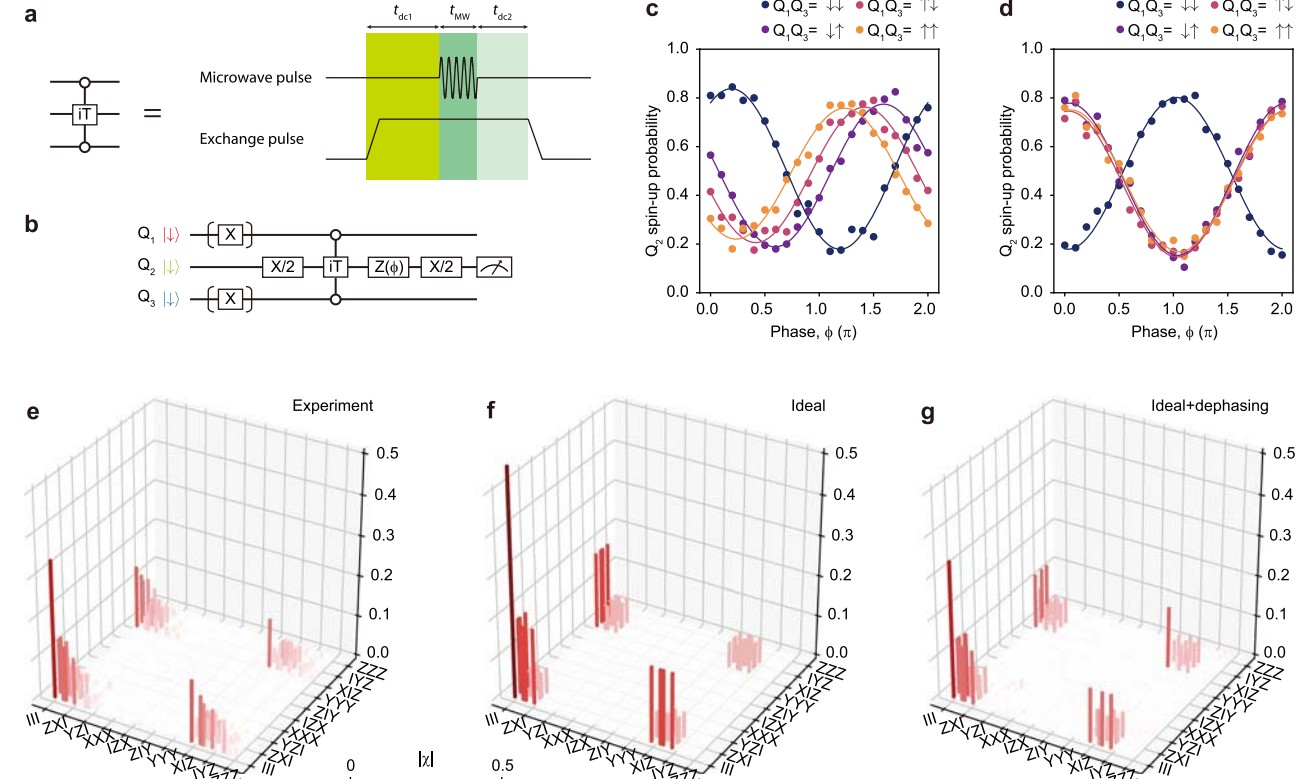

**Extended Data Fig. 5 | iToffoli gate characterizations. a**, Schematic of the iToffoli gate implementation. The iToffoli gate can be realized by a combination of an exchange pulse and a microwave pulse. The exchange pulse duration ($t_{dc1} + t_{MW} + t_{dc2}$), microwave pulse duration ($t_{MW}$) and timing ($t_{dc1} - t_{dc2}$) are fine-tuned to obtain a correct phase evolution. **b**, Quantum circuit used to measure the phase accumulation during the iToffoli gate operation. The iToffoli gate is interleaved between two π/2 pulses to realize Ramsey-type phase detection. Only when $Q_1Q_3 = |{\downarrow}{\downarrow}\rangle$ does a spin flip occur, which is detected as a π phase shift for a correct iToffoli gate. For the other ancilla qubit configurations, the phase accumulation should be zero. **c**, Example phase measurement result before the iToffoli gate phase calibration. The resonance frequency and microwave amplitude are calibrated. **d**, Phase measurement after the calibration of both conditional and unconditional phases. In the calibration procedure, we optimize the duration of the exchange pulse and the timing of the microwave pulse

(see Methods). We obtain correct phase evolution for all ancilla qubit configurations. The phase offsets are $(1.03 \pm 0.01)\pi$, $(0.04 \pm 0.01)\pi$, $(0.03 \pm 0.01)\pi$ and $(0.05 \pm 0.01)\pi$ for $Q_1Q_3 = |{\downarrow}{\downarrow}\rangle$, $|{\uparrow}{\downarrow}\rangle$, $|{\downarrow}{\uparrow}\rangle$ and $|{\uparrow}{\uparrow}\rangle$, respectively. The errors are $1\sigma$ from the mean. **e**, Experimental process matrix ($\chi$ matrix) of the iToffoli gate obtained by three-qubit quantum process tomography (see Methods). The labels represent three-qubit Pauli operators. We obtain a gate fidelity of 0.67 from the data. **f**, Ideal process matrix of iToffoli gate. **g**, Simulated process matrix of iToffoli gate under quasi-static single-qubit phase noise. Here we assume $T_2^* = 1.2$, 1.2 and 1.3 μs for $Q_1$, $Q_2$ and $Q_3$, respectively (ergodic $T_2^*$ measured for long integration time). The effect of charge-noise-induced exchange fluctuation (noise in ZZ term) is not taken into account. The simulation reproduces some features in the experimental data. The gate fidelity estimated from the simulation is 0.69, which agrees well with the experimental result.

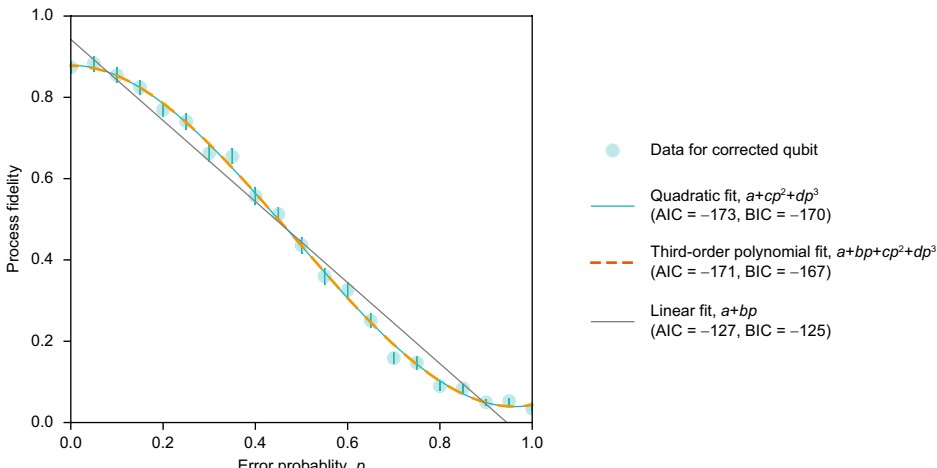

**Extended Data Fig. 6 | Comparison of different models for QEC result.** Comparison of the fitting results of quadratic ($a + cp^2 + dp^3$), third-order polynomial ($a + bp + cp^2 + dp^3$) and linear functions ($a + bp$). The coefficients $a$, $b$, $c$ and $d$ are the fitting parameters. We obtain $a = 0.879 \pm 0.006$, $c = -2.72 \pm 0.06$ and $d = 1.89 \pm 0.06$ (quadratic function), $a = 0.88 \pm 0.01$, $b = 0.0 \pm 0.1$, $c = -2.75 \pm 0.24$ and $d = 1.91 \pm 0.16$ (third-order polynomial function), and $a = 0.94 \pm 0.02$ and $b = -1.00 \pm 0.03$ (linear function) from the fitting results. To compare the fitting results by different models, we calculate the Bayesian information criterion[43] (BIC) and the Akaike information criterion[44] (AIC). Models with lower BIC are preferred, whereas models with lower AIC provide better prediction of the experimental behaviour. The difference between the linear fit and the others is found to be decisive[45] ($|\Delta\text{BIC}| \approx 50$). In addition, although the difference is not as large ($|\Delta\text{BIC}| \approx 3$), the quadratic fit without the first-order term is more preferred than the polynomial function including the first-order term. The errors are $1\sigma$ from the mean.

**Extended Data Table 1 | Evolution of three-qubit state during QEC**

| | $Q_1$ error | $Q_2$ error | $Q_3$ error |
|---|---|---|---|
| Encoded | | $\alpha\left|+++\right\rangle + \beta\left|---\right\rangle$ | |
| Error | $\alpha(\sqrt{1-p}\left|+++\right\rangle + \sqrt{p}\left|-++\right\rangle)$ $+\beta(\sqrt{1-p}\left|---\right\rangle + \sqrt{p}\left|+--\right\rangle)$ | $\alpha(\sqrt{1-p}\left|+++\right\rangle + \sqrt{p}\left|+-+\right\rangle)$ $+\beta(\sqrt{1-p}\left|---\right\rangle + \sqrt{p}\left|-+-\right\rangle)$ | $\alpha(\sqrt{1-p}\left|+++\right\rangle + \sqrt{p}\left|++-\right\rangle)$ $+\beta(\sqrt{1-p}\left|---\right\rangle + \sqrt{p}\left|--+\right\rangle)$ |
| Decoded $(Q_2Q_1Q_3)$ | $(\alpha\left|\downarrow\right\rangle + \beta\left|\uparrow\right\rangle)(\sqrt{1-p}\left|\downarrow\downarrow\right\rangle + \sqrt{p}\left|\uparrow\downarrow\right\rangle)$ | $\sqrt{1-p}(\alpha\left|\downarrow\right\rangle + \beta\left|\uparrow\right\rangle)\left|\downarrow\downarrow\right\rangle$ $+\sqrt{p}(\beta\left|\downarrow\right\rangle + \alpha\left|\uparrow\right\rangle)\left|\uparrow\uparrow\right\rangle$ | $(\alpha\left|\downarrow\right\rangle + \beta\left|\uparrow\right\rangle)(\sqrt{1-p}\left|\downarrow\downarrow\right\rangle + \sqrt{p}\left|\downarrow\uparrow\right\rangle)$ |
| Corrected $(Q_2Q_1Q_3)$ | $(\alpha\left|\downarrow\right\rangle + \beta\left|\uparrow\right\rangle)(\sqrt{1-p}\left|\uparrow\uparrow\right\rangle + \sqrt{p}\left|\downarrow\uparrow\right\rangle)$ | $(\alpha\left|\downarrow\right\rangle + \beta\left|\uparrow\right\rangle)(\sqrt{1-p}\left|\uparrow\uparrow\right\rangle + i\sqrt{p}\left|\downarrow\downarrow\right\rangle)$ | $(\alpha\left|\downarrow\right\rangle + \beta\left|\uparrow\right\rangle)(\sqrt{1-p}\left|\uparrow\uparrow\right\rangle + \sqrt{p}\left|\uparrow\downarrow\right\rangle)$ |
| Error syndrome | $\left|\downarrow\uparrow\right\rangle$ | $\left|\downarrow\downarrow\right\rangle$ | $\left|\uparrow\downarrow\right\rangle$ |

The $Q_2$ input state $\alpha\left|\downarrow\right\rangle + \beta\left|\uparrow\right\rangle$ is encoded to the three-qubit state $\alpha\left|+++\right\rangle + \beta\left|---\right\rangle$. For the decoded and corrected states, we write $Q_2$ first for the sake of brevity. When the error is a coherent phase rotation $Z(\theta)$, the error coefficient $\sqrt{p}$ ($\sqrt{1-p}$) is replaced with $-i\sin(\theta/2)$ ($\cos(\theta/2)$), whereas the result remains essentially equivalent.