## [Peer Review File · Nature]

Manuscript Title: Quantum error correction with silicon spin qubits.

Reviewer Comments & Author Rebuttals

Reviewer Reports on the Initial Version:

Referees' comments:

Referee #1 (Remarks to the Author):

This paper reports a demonstration of quantum error correction (QEC) using spin qubits in silicon, in a 3-qubit device.

There are a number of aspects of this demonstration that make it quite exciting, and worthy of consideration for publication in Nature. First is that this experiment represents the first clear demonstration of QEC in silicon spin qubits, and as these qubits are emerging as a very competitive approach to building quantum computer technology, this demonstration represents a significant milestone in their development. Second, their approach makes use of a type of Toffoli logic gate, and this is quite a unique aspect of this demonstration compared with most other architectures.

I would also like to point out a significant caveat in this demonstration of quantum error correction, which is that the demonstration is inherently not fault-tolerant. Having fault tolerance is not a requirement for QEC, but a fairly standard feature of QEC demonstrations is that the syndrome data is read out classically, and that this allows these data to be processed with classical computers in order to select the correction. In the current demonstration, using a Toffoli gate, they have performed the correction without having (yet) measured the syndrome data. As the decoding problem for this particular code is so simple, they are able to get away with this simplification, but it means that this approach cannot be applied to more sophisticated codes and decoders. As a result, the significance of this demonstration to the ongoing research program is somewhat diminished.

On the other hand, I find that the demonstration and use of the Toffoli-type gate to be a significant result of this paper. Along with this type of QEC demonstration, Toffoli gates have a range of applications in quantum technology and are of independent interest. The characterization of the Toffoli in the diagonal basis is very good, but I would have liked to see its performance on coherent inputs. Can the authors provide a full process matrix for their gate (including phases) given that they undertook quantum process tomography?

Some specific comments:

1. On line 16, it is said that QEC "often involves a three-qubit gate". This goes to my point above; it is not accurate to say that it often involves such a gate, but rather that this Toffoli-type gate is a somewhat unique feature of using this particular code in the limited manner that they have demonstrated. Most QEC schemes, especially fault-tolerant ones, do not use such a gate.
2. The text on lines 46-50 is misleading. The authors are perhaps giving just an example, of one of the syndromes (Z1Z3) and a particular correction based on this syndrome. As it is written, the paragraph suggests this is the entire QEC procedure but rather it is just one aspect of it. I think the authors should improve the presentation of this paragraph to better explain their QEC scheme, and how it relates to more standard approaches involving measurement and feedback.
3. Line 81, there is no justification for why these equatorial qubit states are "the most nontrivial

cases".

4. Figure 4b, I find it very difficult to understand these data and the fits. The authors are suggesting that these fits are giving evidence that the uncorrected case scales linearly and the corrected case scales quadratically, but I find this analysis simplistic and questionable. I also find the data shown in 4d to be fairly inconclusive. Some indication of the uncertainty in these data points would seem necessary, as a minimum, to draw any conclusions.

Because of the concerns above, I do not find this manuscript publishable in its current form.

Referee #2 (Remarks to the Author):

In the manuscript "Quantum Error Correction with Silicon Spin Qubits" the authors claim to have achieved quantum error correction (considered a requirement for any useful gate based quantum computer) in a silicon spins, a first for this promising qubit technology. I find the originality, the presented data, and methods nearly sufficient and appropriate for publication in Nature. In particular, additional information/data to increase the statistical confidence of fig 4's results would strongly bolster its case for publication.

As the authors note, this is not the 1st demonstration of three qubit quantum error correction (QEC), but its accomplishment in a new technology has always been recognized as a prominent milestone in viable qubits technologies like Si spins (most notably superconducting and ions before this, refs 8-9, while NMR in ref 7 is no longer considered viable). The authors lever this context with an approach and structure that is very influenced by refs 8 & 9, which aids credibility and direct comparisons. In particular, the authors likewise employ a Toffoli gate rather than measurement-and-feedback based QEC, and fidelity metrics and presentation are often similar to that in refs 8 and 9. Moreover, credibility of the results also comes from their recent GHZ state demonstration in ref 22 with this same device, which already established the key capabilities necessary for QEC.

I find the evidence for successful QEC to be strongest in figure 3, and based on this data alone I believe that QEC is functional in this system. However, improved performance for the complementary data in fig 4 appears by inspection to be barely at the threshold of statistical significance. In particular, the text claims "the corrected qubit shows improvement...around $p < 0.45$ " for fig 4b, while the improvement barely exceed the marker size in the figure and is possible only for 4 of 9 of the data points in this range. Similarly, while I like that the authors applied QEC in the context of natural dephasing, an even smaller minority of corrected points exceed the uncorrected curve, and a skeptical reader might point out that the uncorrected curve looks to have dropped below its trend line in this region. This is in contrast with the analogous figures in refs 8 & 9: ref 8 features error bars on the un/corrected data points and ref 9 features enough single points that statistical significance can be inferred by inspection. Either of these approaches would strongly bolster the apparent statistical significance of the data in this manuscript.

I also confirm that I find the text of this manuscript sufficiently clear, well-written, and usually well-cited.

Some itemized comments:

Lines 75-79: the authors appear to imply that a bit flip code implementation would have higher fidelity. Is this the case? If so, it seems strange that the authors wouldn't have attempted the cleanest, highest-fidelity approach.

Line 90: cite "...the threshold to witness genuine GHZ-class states."

Line 129: If I understand correctly, a Toffoli gate could also be applied equally well when $Q1Q3 = |uu\rangle$ in this system, and yet the authors first flip these qubits and apply the " $|dd\rangle$ Toffoli" gate. Why?

Lines 155-156: citing improved coherence times and gate fidelities as the path to better performance is so generic as to be meaningless. What is the primary (or top two) error mechanism in this demonstration? Can an error budget be attempted?

Line 330: Why aren't virtual gates applied to the barrier gates?

Fig 4d: Perhaps show the entire $tw = [0,3]$ us curve not as an inset, but as the full graph with a logarithmic tw axis?

Extended Table 1: The qubit ordering is wrong in the 1st column -- should be (Q2,Q1,Q3) not (Q1, Q2, Q3).

Referee #3 (Remarks to the Author):

The manuscript "Quantum error correction with silicon spin qubits" by Takeda et al. demonstrates the implementation of a three-qubit phase flip code in gate-defined Si/SiGe quantum dot spin qubits. The three-qubit phase flip code is an algorithmic code that can correct one phase flip amongst the three qubits, without requiring a measurement during the gate sequence. This makes it particularly interesting for very early quantum error correction demonstrations as shown here.

The manuscript is well written and referenced, and the demonstration of the three-qubit phase flip code is a milestone for any qubit platform. The device performs admirably, as evidenced in Figs. 1 & 2, however these figures do not go beyond what was shown in Takeda et al. Nature Nanotechnology 16, 965 (2021) by the same group. The novelty is in Figs. 3 & 4. Figure 3 shows the proof-of-principle demonstration that the 3-qubit phase flip code is able to correct artificially introduced errors on one of the three qubits, and it does so convincingly. Figure 4 extends this to artificially introduced errors on all three qubits in Fig. 4a and naturally occurring dephasing errors in Fig. 4b. Arguably, this would be the most impactful part of the manuscript, however both measurements are not completely convincing; partly because of the quality of the data and partly because of missing (rigorous) analysis and statistics.

For the three-qubit phase error correction in Fig. 4a, the effect of the corrected sequence is expected to manifest itself in the absence of a linear term. I understand that this is a difficult experiment, however the data, at least as it is currently presented, does not instill confidence that the corrected sequence is able to do so for the following reasons:

1. Some of the data points for the corrected sequence are missing or hidden behind the uncorrected sequence.
2. The presented data points do not have any error bars. Are the fluctuations in the data smaller than the deviation from a linear regression?
3. The fitting parameters are stated without error bars. What is the error on the linear term? Currently it is provided at 0.01, but it could very well be 0.01 ± 0.2 .
4. No statistical method has been employed to demonstrate that the corrected data statistically significantly deviates from the linear regression. Considering errors, can the authors say that the measured data deviates from a linear regression by more than a standard deviation?

In addition, it seems like the artificially introduced error on all three qubits is the same error. So one question that I have here is whether this sort of correlation in error has an effect on the expected outcome of the error correction?

Also, are the corrected and uncorrected sequences of the same length?

For the three-qubit dephasing error correction in Fig. 4b, the effect of the corrected sequence should become obvious from a better state fidelity at short waiting times, where it is likely that only one of the three qubits experiences a phase error. As the authors point out, the initial decay of the fidelity should therefore be less steep. The data presented in Fig. 4b has similar issues as that presented in Fig. 4a and is unconvincing in its current form:

1. The presented data points do not have any error bars.
2. There is no fitting or analysis done on this data. Is there an analytical model that describes the dephasing decay under error correction? When fitted to such model, is there statistically significantly evidence that the error correction leads to an improvement, e.g., slopes different by more than a standard deviation?

Some smaller points:

- L16: A good review paper that talks about scalable manufacturing techniques is Nature Electronics volume 4, pages872–884 (2021).
- L39: For three-qubit entanglement, also <https://arxiv.org/abs/2202.09252> should be cited.
- L115: Not sure if 'classical action' is the best word here. Maybe 'unitary'?
- L132: Adding the word 'intentional' or 'artificially induced' before 'one qubit errors' could make this sentence clearer.
- L133: Similarly, I suggest changing 'The one qubit error is a phase rotation...' to 'We implement the one qubit error as a phase rotation....'.
- L138-142: A sentence that explains the three different curves in Fig. 3b would be great here.
- L282: Are all three microwave sources connected to the same gate?
- Fig. 1a: Why not show the sequence for the three-qubit phase flip code, since this is the one that is demonstrated in this paper?
- L437: 'are used to suppress'
- L456: I would suggest changing 'are always 1' to 'are always high'.
- Extended Data Figure 2: The gate fidelities in f,g,h need to be clearly labelled as primitive gate fidelities. The reader would naturally expect a Clifford gate fidelity here.
- L535: Why is the exponent, and therefore the noise colour, so different for the three qubits?

Author Rebuttals to Initial Comments:

Dear Referees,

We would like to thank you for reviewing our manuscript and for the valuable suggestions and comments to improve the manuscript.

Our point-by-point responses to your comments are given below. All changes in the manuscript (except those of the reference numbers) are highlighted in red. We hope that the revised version of the manuscript addresses all the concerns and is now suitable for publication.

Sincerely yours,

Kenta Takeda

Response to Referee #1:

Comment #1

This paper reports a demonstration of quantum error correction (QEC) using spin qubits in silicon, in a 3-qubit device.

There are a number of aspects of this demonstration that make it quite exciting, and worthy of consideration for publication in Nature. First is that this experiment represents the first clear demonstration of QEC in silicon spin qubits, and as these qubits are emerging as a very competitive approach to building quantum computer technology, this demonstration represents a significant milestone in their development. Second, their approach makes use of a type of Toffoli logic gate, and this is quite a unique aspect of this demonstration compared with most other architectures.

I would also like to point out a significant caveat in this demonstration of quantum error correction, which is that the demonstration is inherently not fault-tolerant. Having fault tolerance is not a requirement for QEC, but a fairly standard feature of QEC demonstrations is that the syndrome data is read out classically, and that this allows these data to be processed with classical computers in order to select the correction. In the current demonstration, using a Toffoli gate, they have performed

the correction without having (yet) measured the syndrome data. As the decoding problem for this particular code is so simple, they are able to get away with this simplification, but it means that this approach cannot be applied to more sophisticated codes and decoders. As a result, the significance of this demonstration to the ongoing research program is somewhat diminished.

Response: We thank the referee for carefully reading our manuscript and finding it worthy of consideration for publication in Nature. As for the concern about syndrome readout, we agree with the referee's comment that the present demonstration has a limited capability for correction, while we argue that our demonstration satisfies the standard for the first realization of QEC in a specific physical system (e.g., refs. 9-11). As in the concluding remarks of our manuscript, we do recognize that the demonstration of a measurement-based protocol is an important step toward fault tolerance and there is some ongoing progress in the field of silicon spin qubit.

Comment #2

On the other hand, I find that the demonstration and use of the Toffoli-type gate to be a significant result of this paper. Along with this type of QEC demonstration, Toffoli gates have a range of applications in quantum technology and are of independent interest. The characterization of the Toffoli in the diagonal basis is very good, but I would have liked to see its performance on coherent inputs. Can the authors provide a full process matrix for their gate (including phases) given that they undertook quantum process tomography?

Response: We thank the referee for finding our iToffoli gate demonstration to be a significant result. In the original manuscript, we showed the data relevant for the demonstrated QEC (diagonal part in Fig. 2h and phase evolution of the data qubit in Extended Data Fig. 5d). As pointed out, the Toffoli-type gate itself has a range of potential applications and our demonstration may stimulate further research in that direction. Therefore, in the revised manuscript, we have included the result of quantum process tomography in Extended Data Fig. 5e-f, where we have obtained a quantum gate fidelity of 0.67. Note that the quasi-static dephasing noise affects stronger for the ancilla qubits because they are undriven during the iToffoli gate. The comparison of the data and simulation indicates that the fidelity is indeed limited by the nuclei-induced quasi-static phase noise. The fidelity is not very good in the present demonstration, but substantial improvement should be possible in an isotopically enriched $^{28}\text{Si}/\text{SiGe}$ device with longer T_2^* times.

Comment #3

1. On line 16, its said that QEC "often involves a three-qubit gate". This goes to my point above; its not accurate to say that it often involves such a gate, but rather that this Toffoli-type gate is a somewhat unique feature of using this particular code in the limited manner that they have demonstrated. Most QEC schemes, especially fault-tolerant ones, do not use such a gate.

Response: We have modified the sentence to represent the use of Toffoli-type gate is a special case as follows:

L19-L20: ... and involves a **three-qubit gate**⁹⁻¹¹ or **measurement-based feedback**, remains an open challenge.

Comment #4

2. The text on lines 46-50 is misleading. The authors are perhaps giving just an example, of one of the syndromes (Z1Z3) and a particular correction based on this syndrome. As its written, the paragraph suggests this is the entire QEC procedure but rather it is just one aspect of it. I think the authors should improve the presentation of this paragraph to better explain their QEC scheme, and how it relates to more standard approaches involving measurement and feedback.

Response: We thank the referee for the suggestion for the improvement of the presentation. The original text tried to describe the specific procedure for the three-qubit code demonstrated in this work, however, it was a bit mixed up with the general QEC scheme. We have modified the sentences to clearly explain the difference between the general QEC and particular QEC schemes used in this work.

L41-L53: **Our three-qubit system (Fig. 1a) comprises one data qubit (Q_2) to be corrected and two ancilla qubits (Q_1 and Q_3). The sequence starts from encoding the data qubit state to a three-qubit entangled state. Then, the phase-flip errors that occurred in the encoded state are mapped to the ancilla qubit states by the decoding. The original data qubit state can finally be restored by a correcting logic gate based on the ancilla qubit states. Most commonly, this correction can be performed by a projective measurement of ancilla qubits followed by a feedback quantum gate on the data qubit. However, this requires a capability to perform high-fidelity qubit measurement much faster than the coherence time, which is still challenging with spins in silicon. While this measurement-based operation is a key component for fault tolerance, in the case of three-qubit QEC, it can alternatively be performed by a multiqubit conditional qubit rotation. In this Article, we take this approach by a three-qubit iToffoli gate, which coherently rotates the data qubit conditioned on the ancilla spin polarization. We synthesize a three-qubit phase-flip code and demonstrate that one qubit phase-flip error can be corrected, and the intrinsic ensemble spin dephasing can be mitigated.**

Comment #5

3. Line 81, there is no justification for why these equatorial qubit states are "the most nontrivial cases".

Response: We have recognized that the original expression ‘nontrivial cases’ was vague in its meaning, therefore we have modified the sentence as follows:

L83-L86: For simplicity, here we perform encoding of an input state on the equator of the Bloch sphere, $Q_2 = (|\downarrow\rangle + e^{i\phi}|\uparrow\rangle)/\sqrt{2}$ (Fig. 2a, ϕ is an azimuthal angle), which is encoded to a maximally entangled three-qubit Greenberger–Horne–Zeilinger (GHZ) state $|\text{GHZ}_\phi\rangle = (|\downarrow\downarrow\downarrow\rangle + e^{i\phi}|\uparrow\uparrow\uparrow\rangle)/\sqrt{2}$.

Comment #6

4. Figure 4b, I find it very difficult to understand these data and the fits. The authors are suggesting that these fits are giving evidence that the uncorrected case scales linearly and the corrected case scales quadratically, but I find this analysis simplistic and questionable. I also find the data shown in 4d to be fairly inconclusive. Some indication of the uncertainty in these data points would seem necessary, as a minimum, to draw any conclusions.

Response: The property tested in Fig. 4b is fundamental to this QEC code (Ref. 14) and demonstrated in the preceding experiments in the other systems (e.g., refs. 10, 11, and 17). One of the reasons that made the interpretation of this data difficult may be the lack of illustration of the ideal behavior in the original manuscript. Now it is included as an inset of Fig. 4b and hopefully, it helps the direct comparison of the experimental data and ideal behavior.

As for the errors, the revised figure has error bars. We agree with the referee that our original claim about quadraticity was too simplistic and lacked appropriate statistical analysis. We have added a discussion about the model selection in Extended Data Fig. 6. We have now included a comparison of the information criterion showing that the quadratic model is likely the best one in the fitting models examined. We have also improved the sentence to be more explicit about what was demonstrated in the experiment.

L157-L160: A quadratic function fits well to the data (see Extended Data Fig. 6 for the comparison between different fitting models). If we allow the first-order term in the fit, it is 0.0 ± 0.1 (the error is 1σ), representing a significant reduction of the first-order sensitivity as compared to the uncorrected case.

Comment #7

Because of the concerns above, I do not find this manuscript publishable in its current form.

Response: We hope that we have addressed all the concerns and the manuscript is now acceptable for publication.

Response to Referee #2:

Comment #1

In the manuscript “Quantum Error Correction with Silicon Spin Qubits” the authors claim to have achieved quantum error correction (considered a requirement for any useful gate based quantum computer) in a silicon spins, a first for this promising qubit technology. I find the originality, the presented data, and methods nearly sufficient and appropriate for publication in Nature. In particular, additional information/data to increase the statistical confidence of fig 4’s results would strongly bolster its case for publication.

As the authors note, this is not the 1st demonstration of three qubit quantum error correction (QEC), but its accomplishment in a new technology has always been recognized as a prominent milestone in viable qubits technologies like Si spins (most notably superconducting and ions before this, refs 8-9, while NMR in ref 7 is no longer considered viable). The authors lever this context with an approach and structure that is very influenced by refs 8 & 9, which aids credibility and direct comparisons. In particular, the authors likewise employ a Toffoli gate rather than measurement-and-feedback based QEC, and fidelity metrics and presentation are often similar to that in refs 8 and 9. Moreover, credibility of the results also comes from their recent GHZ state demonstration in ref 22 with this same device, which already established the key capabilities necessary for QEC.

Response: We thank the referee for carefully reading our manuscript and finding it appropriate for publication in Nature. As for the statistical analysis of Fig. 4, we largely improved it from the original manuscript as in the response to comment #2. We hope that the revised version provides a statistically relevant statement acceptable for publication.

Comment #2

I find the evidence for successful QEC to be strongest in figure 3, and based on this data alone I believe that QEC is functional in this system. However, improved performance for the complementary data in fig 4 appears by inspection to be barely at the threshold of statistical significance. In particular, the text claims “the corrected qubit shows improvement...around $p < 0.45$ ” for fig 4b, while the improvement barely exceed the marker size in the figure and is possible only for 4 of 9 of the data points in this range. Similarly, while I like that the authors applied QEC in the context of natural dephasing, an even smaller minority of corrected points exceed the uncorrected curve, and a skeptical reader might point out that the uncorrected curve looks to have dropped below its trend line in this region. This is in contrast with the analogous figures in refs 8 & 9: ref 8 features error bars on the un/corrected data points and ref 9 features enough single points that statistical significance can be

inferred by inspection. Either of these approaches would strongly bolster the apparent statistical significance of the data in this manuscript.

Response: We thank the referee for the important suggestions. Probably, the most straightforward demonstration is Fig. 3 as pointed out here. The experiment in Fig 4b should work in principle given the demonstration in Fig. 3. Following this and the other referees' comments, we have now included the error bars in the plots to show the statistical significance (Fig. 4b, d). In the original manuscript, the approximate threshold of 0.45 is derived from comparing the data point at $p = 0.45$. As pointed out in the comment, since there is no statistically significant difference between these points, this derivation is not appropriate. In the revised manuscript, we define this threshold by comparing the two fitting curves in Fig. 4b so that the overall behavior of the data is reflected in the threshold value.

L161: ... the corrected qubit shows improvements in a range of $p < 0.429 \pm 0.003$ (the threshold is obtained by comparing the two fit curves in Fig. 4b).

As for the natural dephasing (Fig. 4d), now the readers can recognize that multiple points in the corrected case exceed the data in the uncorrected case by more than one standard deviation.

I also confirm that I find the text of this manuscript sufficiently clear, well-written, and usually well-cited.

Response: We thank the referee for finding our manuscript sufficiently clear, well-written, and well-cited.

Comment #4

Some itemized comments:

Lines 75-79: the authors appear to imply that a bit flip code implementation would have higher fidelity. Is this the case? If so, it seems strange that the authors wouldn't have attempted the cleanest, highest-fidelity approach.

Response: In the manuscript, we do not intend to imply that a bit flip code has a higher fidelity. Since our native two-qubit gate is a CZ gate, it is more natural (and easier) to perform the phase-sensitive encoding/decoding for the phase-flip code. The bit-flip code requires six additional $\pi/2$ rotations (or Hadamard gates) to convert phase-flip to bit-flip. In practice, since the single-qubit gates in our device are fairly reliable, the performance of bit-flip code would be comparable to that of the phase-flip code.

An important difference appears in the demonstration in Fig. 4d, where the errors are the intrinsic noise in the system. Demonstration of intrinsic bit-flip error (T_1 spin relaxation) correction is difficult since the phase coherence would be lost in the relevant time scale ($T_1 > 10$ ms while $T_2 < 0.1$ ms even with spin-echo). For those reasons, we demonstrate the phase-flip code throughout the manuscript.

Comment #5

Line 90: cite "...the threshold to witness genuine GHZ-class states."

Response: We thank the referee for pointing out the lack of citation. We have included a citation about the threshold for witnessing GHZ-class states as follows:

41. Acín, A., Bruß, D., Lewenstein, M. & Sanpera, A. Classification of mixed three-qubit states. *Phys. Rev. Lett.* 87, 40401-1-40401-4 (2001).

Comment #6

Line 129: If I understand correctly, a Toffoli gate could also be applied equally well when $Q1Q3 = |uu\rangle$ in this system, and yet the authors first flip these qubits and apply the " $|dd\rangle$ Toffoli" gate. Why?

Response: As stated in L109-L110, we phenomenologically observe that the $|\downarrow\downarrow\rangle$ Toffoli transition provides the highest contrast Rabi oscillation. Therefore, we utilize the $|\downarrow\downarrow\rangle$ Toffoli gate. While the physical reason remains unclear, a similar observation was reported in a system with two exchange coupled donor electrons (Ref. 35). We also find that there are some other two-qubit demonstrations in Si QDs (Ref. 40 and Huang et al., *Nature* **569**, 532–536 (2019)) that might show state-dependent contrast of the resonance peaks, while there is no statement about the observations in these papers.

Comment #7

Lines 155-156: citing improved coherence times and gate fidelities as the path to better performance is so generic as to be meaningless. What is the primary (or top two) error mechanism in this demonstration? Can an error budget be attempted?

Response: To be more specific, we have clarified that the dephasing during the iToffoli gate is the primary limiting factor in the current experiment.

L163-L164: ..., improvement of the coherence times and thus the fidelity of iToffoli gate, which primarily limits the performance in the corrected case, would ameliorate the situation.

In the other part of the sequence (encoding and decoding), the single-qubit dephasing noise is canceled out by the echo π pulses (echo $T_2 > 10 \mu\text{s}$ instead of $T_2^* = 1 - 2 \mu\text{s}$). Dephasing during the CZ pulses may be the second dominant dephasing source. In addition to dephasing, coherent control errors can cause some performance degradation. For more precise error budgeting, we need (very costly) gate set tomography experiments to obtain full process matrices and error generators of the quantum gates used. The problem of such an experiment may be that the required resource scales exponentially as the number of qubits increases and already is impractical for three-qubit systems.

Comment #8

Line 330: Why aren't virtual gates applied to the barrier gates?

Response: We use the virtual gate technique for our barrier gate pulses as explained in Methods (L353-L357). Since our primary purpose here is to suppress the shift of the energy detuning, we do not apply this technique to compensate for the crosstalk between the exchange couplings. We could technically do so as demonstrated in e.g., Qiao et al., PRX 10, 031006 (2020).

Comment #9

Fig 4d: Perhaps show the entire $tw = [0,3]$ us curve not as an inset, but as the full graph with a logarithmic tw axis?

Response: We thank the referee for this useful suggestion. After preparing the fit curves for the data, we have found that the plots for $tw = [0,3]$ us make much more sense. We now plot all the data in the same plot with a logarithmic time scale.

Comment #10

Extended Table 1: The qubit ordering is wrong in the 1st column -- should be (Q2,Q1,Q3) not (Q1, Q2, Q3).

Response: We thank the referee for pointing out the typo. We have corrected it in the revised manuscript.

Response to Referee #3:

#Comment 1

The manuscript “Quantum error correction with silicon spin qubits” by Takeda et al. demonstrates the implementation of a three-qubit phase flip code in gate-defined Si/SiGe quantum dot spin qubits. The three-qubit phase flip code is an algorithmic code that can correct one phase flip amongst the three qubits, without requiring a measurement during the gate sequence. This makes it particularly interesting for very early quantum error correction demonstrations as shown here.

The manuscript is well written and referenced, and the demonstration of the three-qubit phase flip code is a milestone for any qubit platform. The device performs admirably, as evidenced in Figs. 1 & 2, however these figures do not go beyond what was shown in Takeda et al. Nature Nanotechnology 16, 965 (2021) by the same group. The novelty is in Figs. 3 & 4. Figure 3 shows the proof-of-principle demonstration that the 3-qubit phase flip code is able to correct artificially introduced errors on one of the three qubits, and it does so convincingly. Figure 4 extends this to artificially introduced errors on all three qubits in Fig. 4a and naturally occurring dephasing errors in Fig. 4b. Arguably, this would be the most impactful part of the manuscript, however both measurements are not completely convincing; partly because of the quality of the data and partly because of missing (rigorous) analysis and statistics.

Response: We thank the referee for carefully reading our manuscript. As for Figs. 1&2, the state tomography on GHZ-states is indeed a kind of sanity check and does not go beyond our previous work as pointed out. However, the Toffoli-type gate (Fig. 2e-h) is first demonstrated in silicon, and we think it provides some advances in this field. Regarding the plots in Fig. 4, we admit the lack of rigorous analysis in the original manuscript. We have improved the analysis and statistics in this revision (see the responses for the comments below), and we believe that the revised manuscript is now acceptable for publication.

Comment #2

For the three-qubit phase error correction in Fig. 4a, the effect of the corrected sequence is expected to manifest itself in the absence of a linear term. I understand that this is a difficult experiment, however the data, at least as it is currently presented, does not instil confidence that the corrected sequence is able to do so for the following reasons:

1. Some of the data points for the corrected sequence are missing or hidden behind the uncorrected sequence.

Response: In the original figure, some data points for the corrected case are hidden behind the uncorrected case, while we did plot all the data points. In the revised manuscript, we have made them transparent so that the overlapping points can be recognized.

Comment #4

2. The presented data points do not have any error bars. Are the fluctuations in the data smaller than the deviation from a linear regression?

Response: We include the error bars in the revised manuscript. As for the statistical validity of the model selection, please refer to the response to comment #5 (next one).

Comment #5

3. The fitting parameters are stated without error bars. What is the error on the linear term? Currently it is provided at 0.01, but it could very well be 0.01 ± 0.2 .

Response: We thank the referee for pointing out the important point. The error on the linear term is $0.0(1) \pm 0.1$. Following this and the other referees' comments, we have reconsidered the statement about quadraticity and concluded that even if this error is very small, it would be difficult to claim the quadraticity because there is no relevant threshold for the error. Instead, we have decided to compare the different fitting models based on widely used information criteria (Extended Data Fig. 6). The figure includes a statistical comparison between the quadratic and linear models. We compare the (relative) goodness of the model (Bayesian information criterion (BIC) and Akaike information criterion (AIC)). We have found that the quadratic model is significantly better than the linear model ($|\Delta\text{BIC}| > 50$).

Comment #6

4. No statistical method has been employed to demonstrate that the corrected data statistically significantly deviates from the linear regression. Considering errors, can the authors say that the measured data deviates from a linear regression by more than a standard deviation?

In addition, it seems like the artificially introduced error on all three qubits is the same error. So one question that I have here is whether this sort of correlation in error has an effect on the expected outcome of the error correction?

Response: We believe that the above discussion (reply to comment #5) about the model selection answers the first point. Since the linear fit and quadratic fit cross at some points (see Extended Data Fig. 6), the deviation from the linear fit does not always provide a good criterion to select the model.

Regarding the second point about correlation, the experiment in Fig. 4b is rather a special case where the error is deterministic, and the correlation does not affect the result. Generally, the presence of correlation affects the outcome of this QEC. For example, if the errors are perfectly correlated stochastic full-flip errors on all three qubits (while this situation is extremely unlikely), the QEC always fails because there is no chance of having one error on only one of the three qubits.

Comment #7

Also, are the corrected and uncorrected sequences of the same length?

Response: The answer depends on the definition of ‘sequence length’. If the sequence length means the duration from the first quantum gate to the final quantum gate, it is different between the two cases. In the uncorrected case, we omit the iToffoli gate and apply pre-rotations for tomographic readout directly after the decoding. The interval between the encoding and decoding is the same for both cases. If the sequence length means the total pulse duration relevant for the integration time of data (and T_2^*), it is the same in both cases. In the uncorrected case, we have an additional idle time having the same length as the iToffoli gate after the pre-rotations for state tomography. We have summarized the explanation above in a schematic as follows:

Comment #8

For the three-qubit dephasing error correction in Fig. 4b, the effect of the corrected sequence should become obvious from a better state fidelity at short waiting times, where it is likely that only one of the three qubits experiences a phase error. As the authors point out, the initial decay of the fidelity should therefore be less steep. The data presented in Fig. 4b has similar issues as that presented in Fig. 4a and is unconvincing in its current form:

- 1. The presented data points do not have any error bars.*
- 2. There is no fitting or analysis done on this data. Is there an analytical model that describes the dephasing decay under error correction? When fitted to such model, is there statistically significant evidence that the error correction leads to an improvement, e.g., slopes different by more than a standard deviation?*

Response: We thank the referee for commenting on the problem in Fig. 4. In the revised manuscript, we have included the error bars. It may be possible to compute the dephasing under error correction with knowledge of the three-qubit process matrices of encoding, decoding, and correction and the

noise character of the three qubits. Experimentally, it is not always easy to characterize multiqubit systems in such a way and therefore we adopt an empirical fit function used in the previous study (Ref. 43). The curves are eye guides since they do not describe the underlying physics. As for the slope, it is best practice to directly compare the data point around $t_w = 0 \mu\text{s}$ and the intermediate regime around $t_w = 0.5 - 1 \mu\text{s}$. Initially, the uncorrected fidelity is statistically significantly higher than the corrected fidelity and vice versa for the intermediate regime. While the fitting curves well fit the data and their slope results in the same conclusion, it may not be that definitive due to the unknown physical background of the fit functions.

Comment #9

Some smaller points:

- L16: A good review paper that talks about scalable manufacturing techniques is Nature Electronics volume 4, pages 872–884 (2021).

Response: We thank the referee for suggesting a nice reference for scalable manufacturing. We have added citations as follows (we have also included a recent demonstration of an IBM-made CMOS qubit):

L17: ... silicon-based spin qubits hold a great promise due to their compatibility to mature nanofabrication technologies for scaling up³⁻⁵.

L211-L214:

3. Gonzalez-Zalba, M. F. *et al.* Scaling silicon-based quantum computing using CMOS technology. *Nat. Electron.* **4**, 872–884 (2021).
4. Camenzind, L. C. *et al.* A hole spin qubit in a fin field-effect transistor above 4 kelvin. *Nat. Electron.* **5**, 178–183 (2022).
5. Zwerver, A. M. J. *et al.* Qubits made by advanced semiconductor manufacturing. *Nat. Electron.* **5**, 184–190 (2022).

Comment #10

- L39: For three-qubit entanglement, also <https://arxiv.org/abs/2202.09252> should be cited.

Response: We thank the referee for the suggestion. The 6-qubit paper appeared on arXiv after the initial submission of our manuscript. Of course, it is worth mentioning in our manuscript, and we have included a citation as follows:

L38-L39: ... and generation of three-qubit entanglement^{25,26}.

25. Philips, S. G. J. *et al.* Universal control of a six-qubit quantum processor in silicon. Preprint at <https://arxiv.org/abs/2202.09252> (2022).

Comment #11

- L115: Not sure if ‘classical action’ is the best word here. Maybe ‘unitary’?

Response: We thank the referee for the suggestion. The term ‘classical action’ was used in the preceding report in a superconducting transmon device (Ref. 11), while it may not be common usage now. Since we do not show the full quantum behavior in Fig. 3h, ‘unitary action’ may be misleading. Therefore, we have modified it to be more explicit; ‘gate action on the eigenstates’.

L118-L119: ... represents the experimental (ideal) **gate action on the eigenstates**.

Comment #12

- L132: Adding the word ‘intentional’ or ‘artificially induced’ before ‘one qubit errors’ could make this sentence clearer.

Response: We thank the referee for the suggestion. We have added ‘intentional’ as follows:

L136: ... various **intentional** one qubit errors (see Methods for the details...)

Comment #13

- L133: Similarly, I suggest changing ‘The one qubit error is a phase rotation...’ to ‘We implement the one qubit error as a phase rotation....’.

Response: We thank the referee for the suggestion. We have modified the sentence as suggested.

L137: **We implement the one qubit error as** a phase rotation with a known rotation angle θ , which is equivalent to a phase-flip error with $p = \sin^2(\theta/2)$.

Comment #14

- L138-142: A sentence that explains the three different curves in Fig. 3b would be great here.

Response: We thank the referee for the suggestion. We agree that the original manuscript was lacking a precise description of the experiment. We have added a sentence to explain the experiment as follows:

L141: (corrected Q_i error means a phase-flip error is applied only to Q_i and the correction is performed).

Comment #15

- L282: Are all three microwave sources connected to the same gate?

Response: All three microwave tones are applied to the lower screening gate. We have clarified it as follows:

L316-L317: The outputs of the three signal generators are combined at room temperature and connected to the lower screening gate.

Comment #16

- Fig. 1a: Why not show the sequence for the three-qubit phase flip code, since this is the one that is demonstrated in this paper?

Response: We thank the referee for the suggestion. We have modified the figure for the phase flip code.

Comment #17

- L437: 'are used to suppress'

Response: We thank the referee for correcting the typo. We have implemented the correction.

L471: ... the dashed purple line) are used to suppress the low-frequency single-qubit phase noise.

Comment #18

- L456: I would suggest changing 'are always 1' to 'are always high'.

Response: In the ideal case without any gate errors, it is expected to be exactly one rather than just being high. Since the original sentence was a bit awkward, we have modified it as follows:

L489: ... corrected fidelities are **always equal to one**.

Comment #19

- Extended Data Figure 2: The gate fidelities in f,g,h need to be clearly labelled as primitive gate fidelities. The reader would naturally expect a Clifford gate fidelity here.

Response: We thank the referee for the suggestion. We have modified the labels of Extended Data Fig. 2f-h to make it clear that the measured fidelities are primitive ones.

Comment #20

- L535: Why is the exponent, and therefore the noise colour, so different for the three qubits?

Response: To our knowledge, this has been an open question for some time, and no one has yet answered. For example, in Ref. 7 (our previous measurement in a $^{28}\text{Si}/\text{SiGe}$ DQD), exponents from ~ 1.2 to ~ 1.8 are reported, and in Ref. 26 ($^{28}\text{Si}/\text{SiGe}$ 6QD from the Vandersypen group), exponents from ~ 1.5 to ~ 2.5 are reported. These observations may suggest that the variation originates from charge noise. For instance, an inhomogeneous distribution of charge fluctuators with different switching rates and magnitudes may vary noise color in the relevant frequency range for the echo measurement. However, the present experimental observations are not conclusive enough.

Reviewer Reports on the First Revision:

Referees' comments:

Referee #1 (Remarks to the Author):

Overall, the authors have done an outstanding job at addressing the concerns I raised in my first report. The paper is substantially improved as a result, in my view. Specifically:

1. The framing of the result in the context of past early QEC experiments and also the future pathway to fault-tolerant QEC (which will require measurement and feedback) is now very clear. I appreciate this significant improvement to the presentation.
2. The data presentation and analysis of the key results shown in Fig 4 is now greatly improved. This was important, not just for making the results clear, but also adding confidence that they are seeing the main result that they claim. I particularly appreciate the use of some standard Bayesian/Akaike information criteria to justify the functional dependence of their corrected fidelity (as a function of p), which is quite arguably a superior analysis compared with the previous papers 8 & 9.
3. My other comments were more minor, and have all been addressed to my satisfaction in the new manuscript.

I've also read through the other referees' reports, and note that they had flagged some similar concerns I had, as well as some others. I have checked the changes made to the manuscript in response to these other reports and find them satisfactory.

I believe that the manuscript is now suitable for publication in Nature in its current form. I commend the authors on a very impressive first demonstration of quantum error correction in Si spin qubits.

Referee #2 (Remarks to the Author):

This is the second time I have reviewed "Quantum error correction with silicon spin qubits." I find that the authors have sufficiently addressed my concerns in the revised manuscript. I support the publication of this manuscript in its current form.

Referee #3 (Remarks to the Author):

I thank the authors for considering all comments and carefully implementing the suggested changes. In particular, the analysis and presentation of the data in Fig. 4 has much improved. Together with Ext. Data Fig. 6, Fig. 4 now convincingly shows the effect of the error correction.

There is just one more point that I would like to raise regarding the authors' response to Comment #1: "However, the Toffoli-type gate (Fig. 2e-h) is first demonstrated in silicon, and we think it provides some advances in this field." The Toffoli gate in this manuscript is the first one demonstrated for quantum dots in silicon, but a CCROT-based Toffoli gate has already been demonstrated for donors in silicon in Madzik, Nature 601, 348 (2022).

Overall, I can now recommend publication of this great work in Nature.

With best regards,
Arne Laucht